# Swinging lever mechanism of myosin directly shown by time-resolved cryo-EM

David P. Klebl[1,2,8,9], Sean N. McMillan[2,3,4,9], Cristina Risi[5], Eva Forgacs[5], Betty Virok[5], Jennifer L. Atherton[5], Sarah A. Harris[6], Michele Stofella[2,4], Donald A. Winkelmann[7], Frank Sobott[2,4], Vitold E. Galkin[5], Peter J. Knight[2,4], Stephen P. Muench[1,2,10✉], Charlotte A. Scarff[2,3,10✉] & Howard D. White[5,10✉]

Myosins produce force and movement in cells through interactions with F-actin[1]. Generation of movement is thought to arise through actin-catalysed conversion of myosin from an ATP-generated primed (pre-powerstroke) state to a post-powerstroke state, accompanied by myosin lever swing[2,3]. However, the initial, primed actomyosin state has never been observed, and the mechanism by which actin catalyses myosin ATPase activity is unclear. Here, to address these issues, we performed time-resolved cryogenic electron microscopy (cryo-EM)[4] of a myosin-5 mutant having slow hydrolysis product release[5,6]. Primed actomyosin was predominantly captured 10 ms after mixing primed myosin with F-actin, whereas post-powerstroke actomyosin predominated at 120 ms, with no abundant intermediate states detected. For detailed interpretation, cryo-EM maps were fitted with pseudo-atomic models. Small but critical changes accompany the primed motor binding to actin through its lower 50-kDa subdomain, with the actin-binding cleft open and phosphate release prohibited. Amino-terminal actin interactions with myosin promote rotation of the upper 50-kDa subdomain, closing the actin-binding cleft, and enabling phosphate release. The formation of interactions between the upper 50-kDa subdomain and actin creates the strong-binding interface needed for effective force production. The myosin-5 lever swings through 93°, predominantly along the actin axis, with little twisting. The magnitude of lever swing matches the typical step length of myosin-5 along actin[7]. These time-resolved structures demonstrate the swinging lever mechanism, elucidate structural transitions of the power stroke, and resolve decades of conjecture on how myosins generate movement.

Myosins are molecular motors that move or move along filamentous actin (F-actin)[1]. They perform many functions in eukaryotes, ranging from muscle contraction to organelle transport, with mutations linked to a range of diseases including heart disease, deafness and cancer[1]. Myosins comprise a motor domain, which can be divided into four subdomains (N-terminal, upper 50-kDa (U50), lower 50-kDa (L50) and converter), a light chain-binding domain and a tail region. The converter and light chain-binding domain form the lever, which rectifies and amplifies changes within the motor domain[3].

Hydrolysis of the Mg–ATP chelate (hereafter, ATP) by myosin provides the energy for doing work. In the nucleotide-free state, the myosin motor is strongly bound to F-actin[8,9]. ATP binding opens a cleft between the U50 and L50 domains, reducing the affinity of myosin for F-actin, which dissociates the complex[10]. Once detached, myosin undergoes the recovery stroke, in which the myosin lever becomes primed to generate force, followed by ATP hydrolysis to ADP and inorganic phosphate ($P_i$)[11]. Release of the $P_i$ from myosin in the absence of interactions with actin is slow, precedes release of ADP, and thus limits the rate of energy release. Primed myosin, with ADP and $P_i$ bound, rebinds F-actin, leading to $P_i$ release, cleft closure and generation of movement, proposedly through swinging of the lever (power stroke)[12,13], towards the barbed end of F-actin (+actin)[13] for most myosins. Actin accelerates $P_i$ release up to 1,000-fold. The order in which $P_i$ release, cleft closure and power stroke occur is debated[3,14]. Release of ADP from the complex is, in some myosins, coupled to a second, smaller swing of the lever that completes the structural cycle[15,16]. The current understanding of force generation by myosin motors, based on the structural states available so far, is reviewed in ref. 3.

[1]School of Biomedical Sciences, Faculty of Biological Sciences, University of Leeds, Leeds, UK. [2]Astbury Centre for Structural Molecular Biology, University of Leeds, Leeds, UK. [3]Discovery and Translational Science Department, Leeds Institute of Cardiovascular and Metabolic Medicine, School of Medicine, Faculty of Medicine and Health, University of Leeds, Leeds, UK. [4]School of Molecular and Cellular Biology, Faculty of Biological Sciences, University of Leeds, Leeds, UK. [5]Department of Biomedical and Translational Sciences, Macon & Joan Brock Virginia Health Sciences at Old Dominion University, Norfolk, VA, USA. [6]School of Mathematical and Physical Sciences, University of Sheffield, Sheffield, UK. [7]Department of Pathology & Laboratory Medicine, Robert Wood Johnson Medical School, Rutgers University, Piscataway, NJ, USA. [8]Present address: Department of Cell and Virus Structure, Max Planck Institute of Biochemistry, Martinsried, Germany. [9]These authors contributed equally: David P. Klebl, Sean N. McMillan. [10]These authors jointly supervised this work: Stephen P. Muench, Charlotte A. Scarff, Howard D. White. ✉e-mail: s.p.muench@leeds.ac.uk; c.a.scarff@leeds.ac.uk; whitehd@ODU.edu

The mechanisms of force generation and actin activation of myosin ATPase activity remain controversial, in part owing to a lack of structural information on how myosin initially interacts with actin in its primed state[3,17]. Actomyosin structures in the ADP and nucleotide-free states, obtained by cryo-EM, reveal the architecture of strongly bound acto-myosin complexes in which both the U50 and L50 subdomains interact with actin, the cleft is closed and the lever adopts a post-powerstroke (postPS) position[18]. The structure of the myosin motor in the primed state in the absence of actin, with ADP–$P_i$ or analogues in the nucleotide-binding site, has been solved by X-ray crystallography for multiple myosin classes including myosin-2 (ref. 19), myosin-5 (ref. 15) and myosin-6 (ref. 20). The myosin primed-state structures show an open cleft between the U50 and L50 subdomains, and a primed lever[15]. However, previous attempts to image its attachment to actin have failed. Thus, the structural transitions of the power stroke were unknown until now.

At a steady state of actomyosin ATPase activity, attached primed myosin is rare because it is a weakly bound state that rapidly transitions to a postPS strongly bound state. Thus, the traditional high-resolution structural methods (X-ray crystallography and cryo-EM plunge-freezing approaches) are unable to capture a primed actomyosin structure. Here we have overcome these difficulties by using a myosin-5 mutant construct with higher affinity for actin[6] and an increased lifetime of the attached primed state[5], and by using a microspray method for cryo-EM specimen preparation[4] that permits millisecond time resolution. We thereby obtained a structure of primed actomyosin at 4.4 Å global resolution, which reveals how the myosin motor interacts with F-actin in its primed state to initiate force generation. Through comparison with the corresponding postPS state that primed actomyosin rapidly transitions to, we reveal the structural transitions of the power stroke and directly demonstrate the swinging lever mechanism.

## Trapping primed actomyosin

To trap the primed actomyosin complex, we pre-incubated a myosin-5 construct (motor domain plus one IQ light chain-binding domain) with ATP for about 2 s, allowing the myosin to bind and hydrolyse ATP, so it was primed for actin binding[5]. This was then mixed rapidly with F-actin, sprayed onto an EM grid and plunge-frozen to trap the reaction after 10 or 120 ms using our custom-built device[4,21] (Methods and Supplementary Fig. 1). We used a myosin-5 mutant with an S217A substitution in switch 1 in the nucleotide-binding pocket and DDEK(594–597) deletion in loop 2 (Supplementary Fig. 2). S217A slows actin-activated $P_i$ release (198 $s^{-1}$ to 16 $s^{-1}$)[5], and the deletion increases the affinity of the myosin-5–ADP–$P_i$ primed state for F-actin about tenfold[6]. This double-mutant motor is fully functional in actin-motility assays and has a maximum actin-activated $P_i$-release rate of 13 $s^{-1}$ (Supplementary Fig. 3).

We chose two time points at which to vitrify the mixture of myosin and actin, 10 and 120 ms. At 10 ms, the maximum speed of the set-up, based on the kinetic data, we expected most actomyosin complexes to still be in the primed state, whereas at 120 ms, a higher proportion of these would have transitioned to a postPS state, ensuring that any intermediate states between the primed and postPS could be captured (see Supplementary Fig. 3).

The time-resolved cryo-EM data at both time points yielded two distinct classes of actomyosin-5 structures, which we identified as the primed and postPS states, and solved to global resolutions of 4.4 and 4.2 Å, respectively (Supplementary Fig. 4). To enable detailed interpretation of the cryo-EM density maps, we followed current practice[22–25] to create pseudo-atomic models by flexibly fitting available high-resolution structures into the maps, performing homology modelling and refinement and using molecular dynamics (MD) simulation approaches to explore side-chain interactions at contact interfaces (Methods and Extended Data Tables 1 and 2). Calmodulin density in all of the EM maps is weak, indicating low occupancy of the heavy chain by calmodulin. The postPS actomyosin structure was similar to previous structures of strongly bound states of myosin-5 (ref. 18), as expected given the mutant we used is fully functional in motility assays and only has a twofold increase in ADP release rate compared to the wild-type motor[5]. Intermediate states between the primed and postPS states were not detected despite extensive three-dimensional classification and masking (Supplementary Fig. 4). This suggests that the lever swing mechanism may be a continuum structural change, analogous to firing a Roman catapult, but intermediate states present in low abundance could have escaped detection. As expected, rigor actomyosin was not identified in the time-resolved data as the high concentration of ATP in the mixture meant that ATP rebinding and dissociation of myosin–ATP from actin would occur rapidly.

The lever swing mechanism predicts that following mixing of primed myosin with F-actin, primed actomyosin will initially predominate, with postPS actomyosin accumulating over time. We found that 62% of actomyosin complexes were in the primed state at 10 ms (Extended Data Fig. 1). At 120 ms, the proportion of primed actomyosin complexes was reduced to 36%, concomitant with an increase in postPS complexes, in reasonable agreement with a $P_i$-release rate of 13 $s^{-1}$ (Supplementary Fig. 3d). This time dependence of conformation directly demonstrates the swinging lever mechanism.

## Structure of primed actomyosin

In the primed state, myosin interacts with actin through its L50 domain with its actin-binding cleft open (Fig. 1 and Supplementary Video 1). The central actomyosin interface is formed between two neighbouring actin subunits and the myosin helix–loop–helix (HLH) motif (Fig. 1a–d), with additional interactions between actin and myosin loop 2 and myosin loop 3 completing the interface (Fig. 1d). The main contacts are primarily hydrophobic in nature, supplemented by electrostatic interactions. The interactions between the HLH and actin and those between loop 3 and actin are the same as observed for the strong-binding states[26], largely conserved across myosin classes in higher eukaryotes[27]. Thus, the orientation of the primed motor domain when docked onto actin resembles that of strongly bound states except that the U50 does not interact with actin in primed actomyosin (Fig. 1b).

Myosin loop 2 is flexible and poorly resolved in the primed state, as in other actomyosin structures[18,26]. Yet, the carboxy-terminal portion of loop 2 (residues 628–632) has appreciable density that adopts an elongated conformation, reaching out parallel to the actin surface. MD-driven fitting and subsequent simulation of side-chain interactions suggest that the positively charged residues K629 and K632 interact with the negatively charged D24 and D25, respectively, in actin subdomain-1 (Fig. 1e, Extended Data Fig. 2 and Extended Data Table 2). A ridge of weaker density extends further along the surface of the actin, suggesting that more of loop 2 may be associated with the actin surface.

The converter is in a primed position within the motor domain, and the orientation of the motor domain on actin results in the emerging lever helix pointing along the actin axis towards the pointed end, at an angle of about 52° to the actin axis (Fig. 1f).

The N-terminal residues of actin (residues 1–4, DEDE for skeletal α-actin used here, and –DDD and –EEE for β-actin and γ-actin, respectively), which are unresolved in most actin structures, can reach out to interact with helix W of the myosin L50 subdomain and loop 2. Modelling in ISOLDE showed that the actin residues D1 and E2 can interact with H637 and N641 in helix W, respectively, and the acetyl group of the acetylated N-terminal residue D1 can interact with H631 in loop 2 (Fig. 1g). Further examination of these modelled interactions, by explicit solvent MD simulations, indicated that they are largely maintained over time (Extended Data Fig. 2 and Extended Data Table 2). These N-terminal actin interactions with myosin may thus lead to the subtle changes in primed myosin structure, described below, and suggest how actin activates myosin ATPase activity.

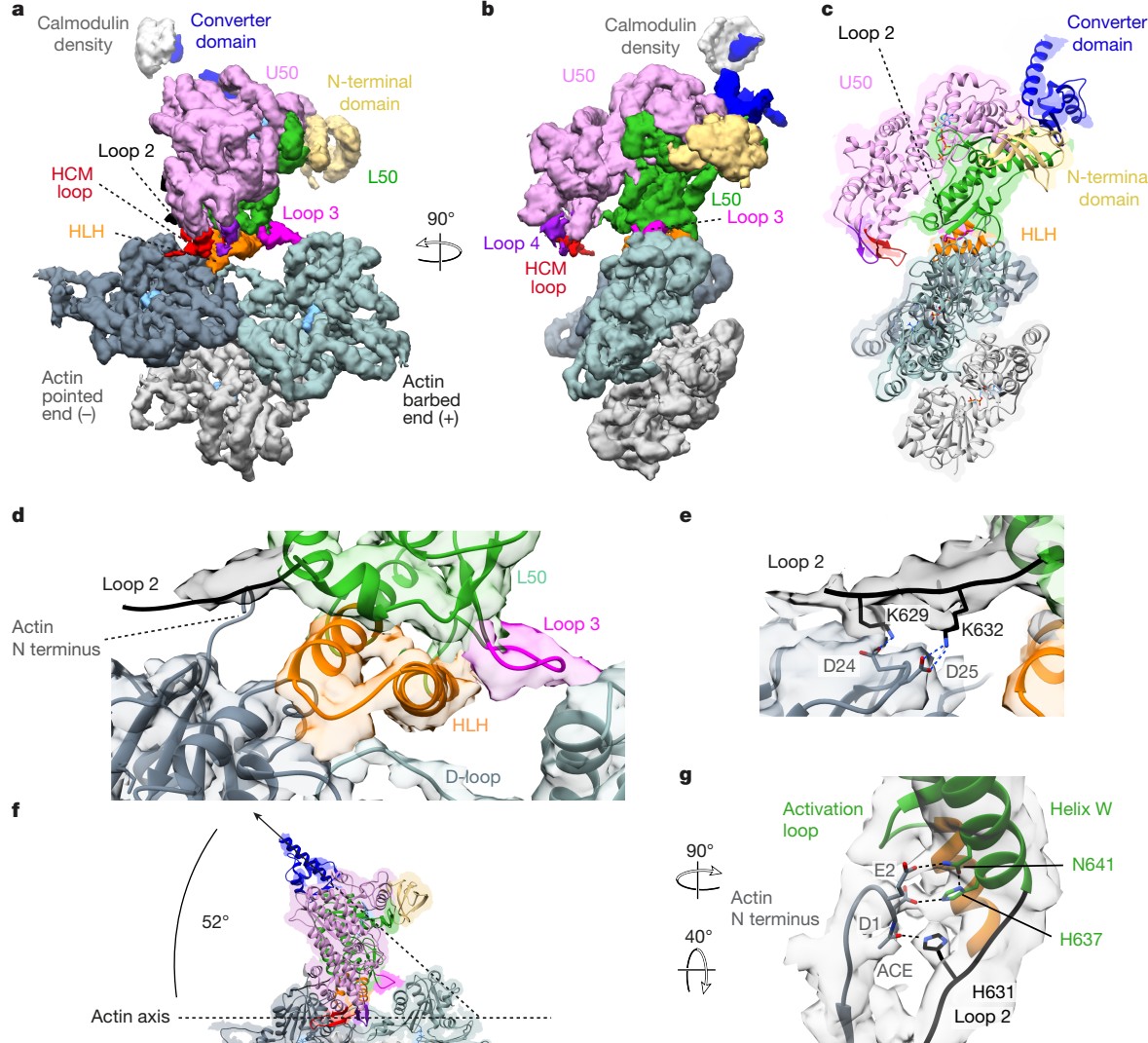

**Fig. 1 | Structure of the primed actomyosin-5 complex. a,b,** Cryo-EM density map of the primed actomyosin-5 complex, segmented and coloured by myosin subdomains and actin chains as indicated (with the central three actin subunits displayed). Actin subunits are shown in slate grey (−actin, nearer the filament pointed (−) end), blue-grey (+actin, nearer the barbed (+) end) and light grey; the nucleotide is shown in sky blue. The map is thresholded to show secondary structure (myosin 0.085, actin 0.2) and is shown in side view of F-actin (**a**) and in end-on view of F-actin, looking towards the pointed end (**b**). **c,** Backbone depiction of a pseudo-atomic model of primed actomyosin-5, fitted into the EM density map, viewed as in **b**. **d,** Magnified side view of the actomyosin interface; the main contacts are made by the myosin HLH motif and loop 3, as observed in strongly bound states. **e,** Additional contacts are made by loop 2 (EM density threshold 0.007). Relevant interacting residues are labelled and shown. **f,** The lever helix points along the actin axis towards the pointed end, at an angle of about 52° to the actin axis. **g,** Magnified view showing the N-terminal residues (D1 and E2) of the −actin subunit (slate grey), interacting with helix W (H637 and N641) of the L50 domain, and loop 2 (H631) (EM density threshold 0.007). A DeepEMhancer post-processed map is depicted in **a**–**d,f,** and a RELION post-processed map is depicted in **e,g**.

## Structural changes following actin binding

The time-resolved cryo-EM data contained unbound myosin-5 molecules, providing us with the opportunity to directly compare myosin structure in the unbound and actomyosin states (Fig. 2 and Supplementary Video 2). Unbound myosin motors from the 120-ms data were analysed to produce an EM map with global resolution of 4.9 Å (Supplementary Fig. 5). This revealed that unbound myosin motors were in a primed state, vitrified before productive actin binding. The crystal structure of the myosin-5c motor domain trapped in the primed state by use of ADP–vanadate (Protein Data Bank ID 4zg4) was well accommodated within the cryo-EM density[15], except in the position of the converter domain and relay helix (Extended Data Fig. 3a–e). Thus, flexible fitting of this crystal structure in the map was used to produce a model of our myosin-5a construct in the unbound primed state (Extended Data Fig. 3a,c,e).

The myosin models for unbound primed myosin-5 and primed actomyosin-5 are very similar (0.80 Å root mean squared deviation from global alignment of the motor domains across 708 Cα atom pairs) yet subtle changes are seen in the flexible regions, especially in the position of the converter domain and helix D (Extended Data Fig. 3f,g). When the two structures are aligned on the main actin-binding interface (the HLH (residues 505–530) alone (Fig. 2a)), the entire U50 is observed to be displaced with the largest shift in the position of helix D (Fig. 2b and Supplementary Video 2). This suggests that subtle structural changes in the myosin motor are induced by actin binding and propagated through the molecule.

In the bound state, with myosin anchored to actin through the HLH, the rest of the L50 moves downwards, relative to the actin axis, so that the U50 domain is rotated circumferentially around F-actin towards the converter, which lifts the HCM loop and loop 4 further away from the actin surface (Fig. 2a,c and Supplementary Video 2). The interaction

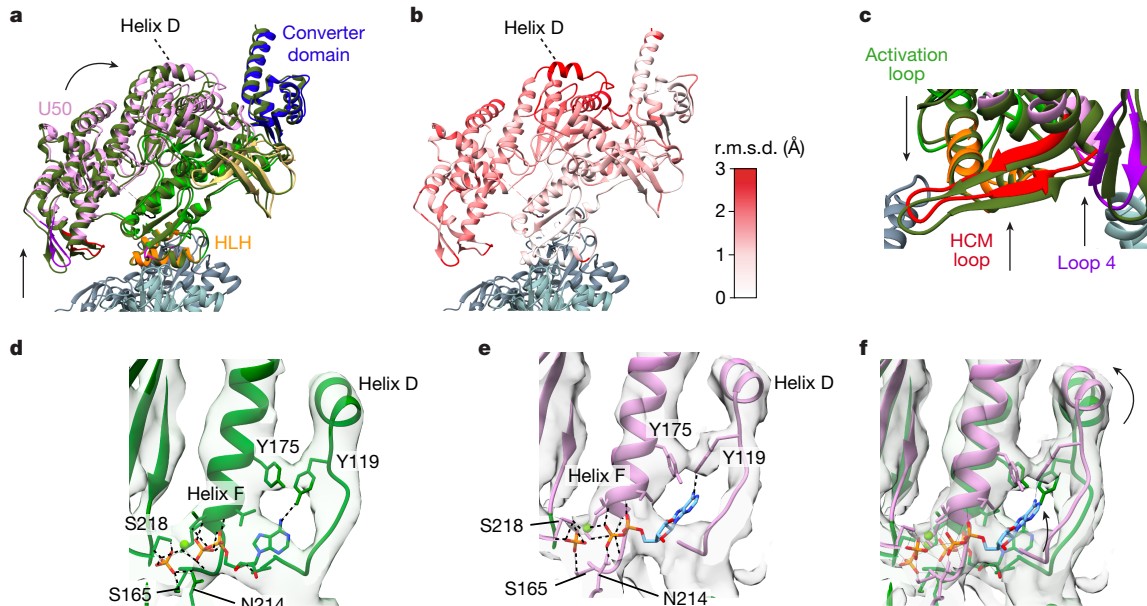

**Fig. 2 | Comparison of myosin structure in the primed actomyosin complex with unbound primed myosin. a**, Superposition of the primed actomyosin (coloured as in Fig. 1) and unbound primed myosin (forest green) aligned on the core primed actomyosin interface (HLH motif, residues 505–530). View towards actin pointed end. **b**, Corresponding root mean squared deviation (r.m.s.d.) of myosin residues between primed actomyosin and primed myosin, showing that the greatest movement occurs in helix D. **c**, The whole U50 is rocked back, around the actin axis, towards the converter domain, resulting in the HCM loop and loop 4 moving away from the actin surface. The activation loop also extends down, reaching out to the actin surface. **d,e**, Unbound primed myosin (**d**) and primed actomyosin (**e**) models focused on helix D, Y119, Y175 and nucleotide, overlaid with their respective cryo-EM maps, thresholded equivalently. **f**, Overlay of **d,e** showing movement of helix D following binding of myosin to actin causes rearrangement of the tyrosine residues Y119 and Y175, resulting in larger freedom of placement of ADP in the nucleotide-binding pocket. The $P_i$ is anchored by interactions with P-loop (S165), helix F (K169) and switch 1 (N214 and S218). A primed actomyosin RELION post-processed map is depicted throughout.

of the N-terminal residues (1–2) of actin with the neighbouring helix W and loop 2 (Fig. 1g) could drive this motion. The C-terminal end of loop 2 extends to contact the actin surface (Fig. 1e), and the 'activation loop' (residues 501–504, between helix Q and helix R) protrudes further out from the axis of its neighbouring helices, reaching out for the actin surface (Fig. 2c). Notably, the converter hardly moves relative to the HLH motif (Fig. 2b), such that there is little movement of the lever when primed myosin binds to actin.

The movement of helix D results in the rearrangement of the position of Y175 and Y119 (Fig. 2d–f and Supplementary Video 2). These residues interact with the adenosine ring of the nucleotide and the rearrangement probably results in less restraint on ADP in the nucleotide pocket. This is supported by the observation of weaker density for the adenosine ring in the primed actomyosin EM density in comparison to that in the unbound primed myosin EM density (Fig. 2d,e). Consequently, in the primed actomyosin model, refined with MD in ISOLDE[28], the ADP is placed further back into the pocket, towards helix D, creating strain that may promote $P_i$ dissociation from the ADP moiety as the $P_i$ is anchored by interactions with the P-loop (S165), helix F (K169) and switch 1 (N214 and S218; Fig. 2d–f). However, $P_i$ cannot dissociate because its exit route is blocked. The loops of switch 1 and switch 2 are in such close proximity in the density map that in our model the salt bridge between R219 in switch 1 and E442 in switch 2, termed the backdoor, is still present and so the exit door is closed.

## Structural changes in the power stroke

Comparison of the primed and postPS actomyosin states allows us to describe the structural changes that occur during the power stroke that could never before be described. The biggest change is the large-scale movement of the converter and light chain-binding domain (Fig. 3a–e, Extended Data Fig. 4 and Supplementary Video 3), responsible for generation of external mechanical force. The lever swings through about

93°, predominantly along the actin axis, and is displaced azimuthally by only 4° right-handed (Fig. 3c–e), with a small (2.5°) right-handed torsion of the lever around its own axis. The N-terminal domain is displaced by about 10 Å orthogonally to the actin axis (Fig. 3e). Thus, the myosin-5 motor successfully converts complex internal movements into a simple swinging motion along actin. These data directly demonstrate the structural characteristics of the swinging lever mechanism.

Whereas the interactions between actin and the myosin L50 domain (HLH motif and loop 3) remain largely unchanged between the primed and postPS states, the U50 interactions are distinctly different (Fig. 3f,g). In the primed state, the HCM loop and loop 4 are poorly resolved, indicating flexibility in this region, and both loops are too distant from the actin surface to form stable contacts with it (Fig. 3f). In the postPS state, the U50 domain is rotated such that the actin-binding cleft is closed and the HCM loop and loop 4 can interact with the actin surface, forming both hydrophobic and charged interactions (Fig. 3g and Extended Data Fig. 5a–d), as seen in previous strongly bound actomyosin structures[18,26,27]. These additional interactions increase the surface area of the binding interface from 375 Å² in the primed state to 729 Å² in the postPS state, creating a much stronger binding interface and providing the structural basis for the transition from weak to strong binding.

In the postPS actomyosin structure, the interactions of loop 2 with actin subdomain 1 are different to those seen in the primed structure (Fig. 3h,i). Pseudo-atomic modelling suggests that interactions of H631 with acetyl-D1 and K629 with D24 (Fig. 3h) are broken and the C-terminal portion of loop 2 adopts a helical conformation with K629 and K633 forming stronger ionic interactions with actin E2 (Fig. 3i). The preserved interaction of K632 in loop 2 with D25 means that the change in loop 2 conformation, which shortens loop 2, would rotate the U50 around towards the actin surface, resulting in formation of the second binding interface and cleft closure (Supplementary Video 3). An interaction between residue K502 in the activation loop and

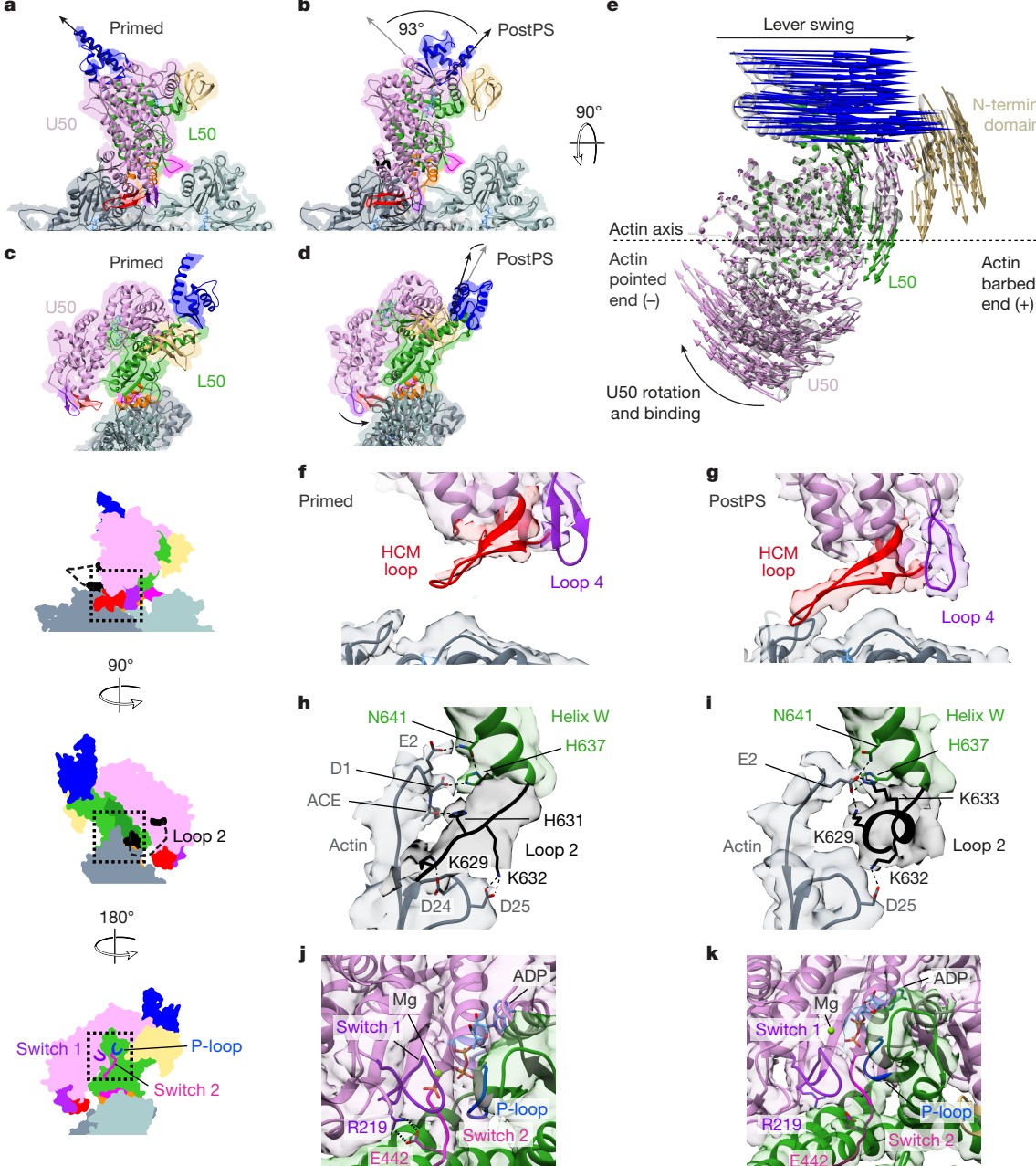

**Fig. 3 | Structural changes during the power stroke. a**,**b**, Primed actomyosin structure (as shown in Fig. 1a; **a**) and the corresponding view of the postPS actomyosin structure (**b**) with lever positions indicated by a black arrow. The lever swings about 93° between structures. **c**,**d**, In the end-on view, primed actomyosin is observed to have an open actin-binding cleft (**c**), whereas postPS actomyosin has a closed cleft (**d**). **e**, In the top view, vectors depict the movement of myosin residue Cα atoms between primed and postPS actomyosin states. The biggest motions are attributable to lever swing, U50 rotation and binding to actin, and movement of the N-terminal domain. Schematic representations are shown of primed actomyosin in side view (to the left of **f**), end-on view towards the barbed end of F-actin (left of **h**) and end-on view towards the pointed end of F-actin (left of **j**), with dashed boxes illustrating the area of the structure highlighted in **f** and **g**, **h** and **i**, and **j** and **k**, respectively. **f**,**g**, The HCM loop and loop 4 are distant from the actin surface in the primed state (**f**) but interact with actin in the postPS state (**g**). The EM density is segmented and coloured by myosin subdomain (contour level 0.008). **h**,**i**, N-terminal actin interactions with loop 2 and helix W are changed between primed (**h**) and postPS (**i**) states. **j**,**k**, Nucleotide-binding site in primed (**j**) and postPS (**k**) structures. The EM density is segmented and coloured by myosin subdomain (contour level primed, 0.0085; postPS, 0.18). The backdoor (salt bridge between R219 and E442) is opened through rotation of the U50 and switch 1 and P-loop moving away from switch 2 (see Supplementary Video 3 and EM density maps). A DeepEMhancer post-processed map is depicted in **a**–**d**,**f**,**g**), and a RELION post-processed map is depicted in **h**–**k**.

E4 of actin can also form only in the postPS state (Extended Data Figs. 2 and 5e and Extended Data Table 2).

Within the nucleotide-binding pocket, there are relative movements between switch 2, switch 1 and the P-loop that indicate that $P_i$ has been released in the postPS structure (Fig. 3j,k). The salt bridge between R219 in switch 1 and E442 in switch 2 (termed the backdoor) is intact in our primed actomyosin pseudo-atomic model and broken in our postPS model, consistent with previous structural observations[3,15,18].

Two possible escape routes for $P_i$ from the active site have previously been proposed, through opening of the backdoor (either by switch 1 moving away from switch 2 (ref. 29) or by switch 2 moving away from switch 1 (ref. 20)) or through formation of a gap between switch 1 and

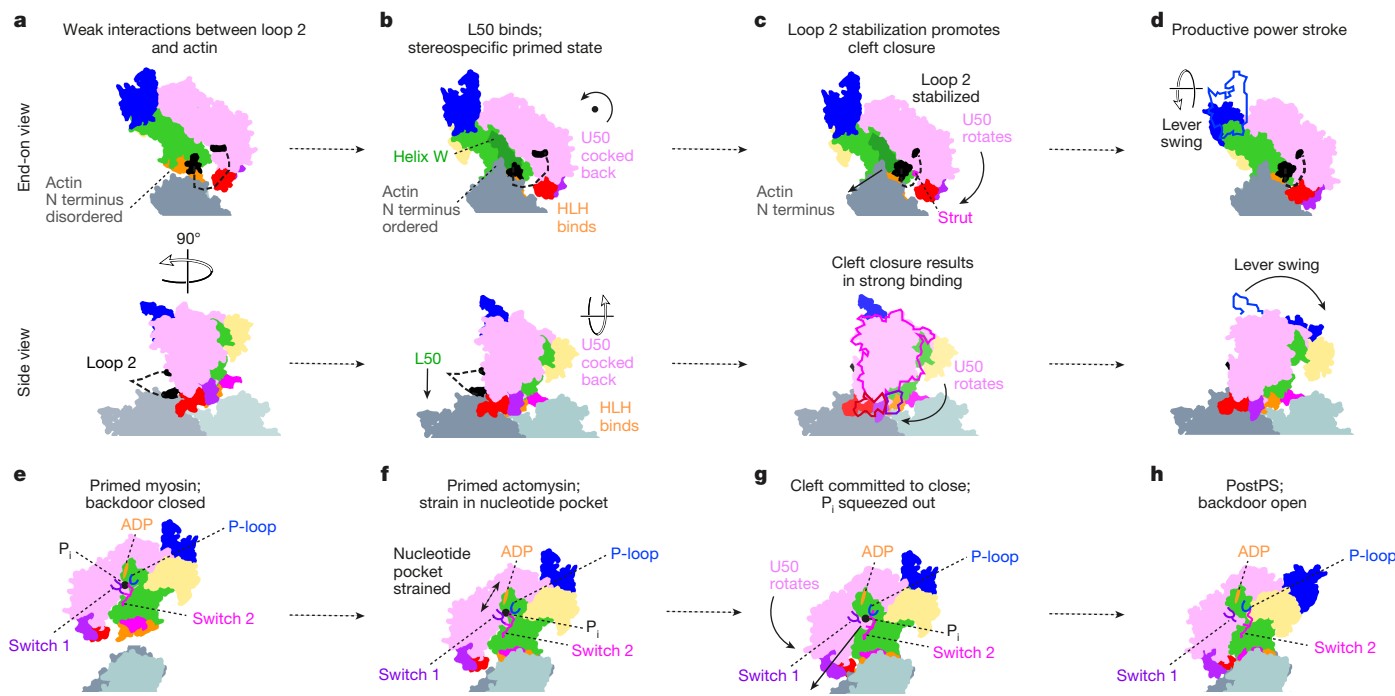

**Fig. 4 | Models of myosin force generation and ATPase activation on F-actin.**
**a**–**d**, Force generation (upper row: end-on view; lower row: side view). **e**–**h**, ATPase activation. **a**, Primed myosin initially binds weakly to actin through electrostatic interactions of loop 2 with actin subdomain 1. This brings the L50 of myosin near to the actin surface, enabling formation of the stereospecific primed actomyosin state. **b**, HLH binding enables the actin N-terminal residues 1–2 to interact with helix W and loop 2, resulting in the U50 being cocked back towards the converter domain, rotated around the F-actin axis. **c**, Rearrangement of N-terminal actin interactions with helix W and loop 2 results in loop 2 stabilization at its C-terminal end. This shortens loop 2, rotating the U50 and attracting the negatively charged strut to positively charged loop 2, promoting cleft closure. **d**, Cleft closure results in the strong-binding interface needed to sustain force

and concomitantly results in twisting of the transducer, straightening of the relay helix and lever swing. **e**, In the unbound primed state, the backdoor is closed, prohibiting $P_i$ release. **f**, Following binding of primed myosin to actin, cocking back of the U50 towards the converter creates strain in the nucleotide pocket, with the ADP drawn away from the well-coordinated $P_i$, prohibiting reversal of hydrolysis and promoting $P_i$ release. **g**, As the U50 rotates, and the initial interactions between the U50 and the actin surface are formed, switch 1 and the P-loop are displaced relative to switch 2, the backdoor is opened, and $P_i$ is squeezed out into the $P_i$ release tunnel. **h**, In the postPS state, $P_i$ has been released, the lever has swung and the backdoor is open. $P_i$ re-entry into the nucleotide pocket is highly unfavourable.

the P-loop[30] (so-called side-door). Here, by comparison of the primed and postPS actomyosin states, we definitively show that rotation of the U50 across the L50, resulting in cleft closure, displaces switch 1 and the P-loop away from switch 2 to open the backdoor and enable $P_i$ release (see morph between primed and postPS actomyosin in Supplementary Video 3).

Our postPS actomyosin-5 structure shows a closed actin-binding cleft as well as a postPS lever position (Fig. 3d) and has high similarity to previous structures of strongly bound actomyosin complexes (ADP-bound or rigor states)[18]. The cryo-EM density shows clear evidence for the presence of MgADP (Extended Data Fig. 5f) and we, therefore, identify the postPS state as ADP-bound actomyosin-5. We observe that the position of the lever is more similar to that observed in previous rigor structures (Extended Data Fig. 5g), rather than ADP-bound structures[15,18]. We also observe that the density for the magnesium ion in the nucleotide-binding pocket (Extended Data Fig. 5h,i) is in a different position to that seen in other ADP-bound structures[18]. This could be due to the S217A substitution changing the Mg coordination, and may explain the twofold increase in ADP release rate for the S217A mutant compared to the wild type[5], along with the change in lever position. To ensure that this mutant motor still transitioned through the canonical mechanochemical cycle, with the expected second smaller lever displacement seen following ADP release[18], we also obtained a cryo-EM density map of the mutant rigor actomyosin to a global resolution of 3.9 Å (Extended Data Fig. 6). This showed the expected lever and converter displacement on ADP release (Extended Data Fig. 6), indicating that this motor undergoes

a classic mechanochemical cycle. The four structures we present thus delineate the entire structural sequence of the myosin-5 power stroke.

## Structural basis of force generation

The changes we observe between unbound primed myosin, primed actomyosin and postPS actomyosin provide many insights that allow us to propose the mechanism by which myosin generates movement and actin catalyses it (Fig. 4 and Supplementary Video 4).

It is generally accepted that myosin initially binds weakly to actin through interactions between positively charged residues of loop 2 and negatively charged residues in actin subdomain-1 (ref. 26; Fig. 4a), which are indeed seen in our primed structure (Fig. 1f). This brings the L50 in close proximity to the actin surface, enabling the stereospecific interaction between the L50 (HLH and loop 3) and F-actin to form quickly after this initial interaction. This interaction triggers an important rearrangement within the primed myosin (cocking back of the U50) to produce the primed actomyosin we observe (Fig. 4b). During the transition between primed and postPS states, we show that the U50 must rotate, resulting in cleft closure and producing the strong-binding interface required to sustain the force generated by lever swing (Fig. 4c,d). Yet, the question of how actin activates myosin ATPase activity still remains.

Actin N-terminal residues 1–4 are implicated in myosin ATPase activation, as deletion or alteration of these residues diminishes actin-activated ATPase activity[31,32]. We find that actin structure is almost

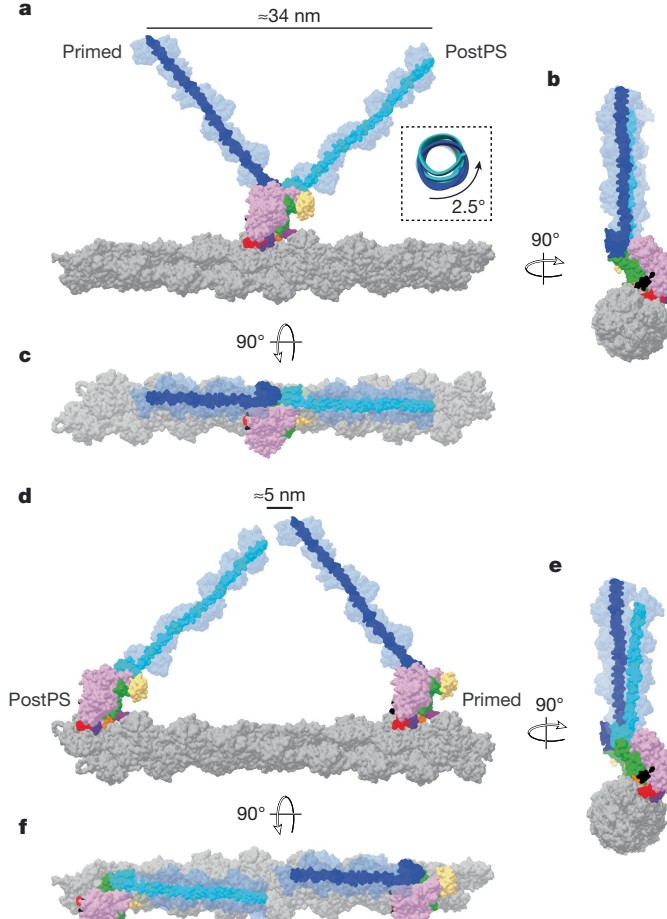

**Fig. 5 | Myosin-5 working stroke and walking on F-actin. a**, Overlay of primed and postPS actomyosin structures with full-length levers, coloured in dark blue and cyan, respectively, on actin in side view. A working stroke of approximately 34 nm is seen as well as little rotation of the lever, as highlighted. **b,c**, End-on (**b**) and top (**c**) views of the actin filament show a very small azimuthal displacement of the lever tips (7°). **d–f**, When a postPS and a primed myosin are positioned 13 actin subunits apart, the two lever ends are close, as observed in side (**d**), end-on (**e**) and top (**f**) views of the actin filament. Note that this actin filament has a rotation per subunit of −166.6°. Small changes in this value change the relationship of the lever ends in **d–f**.

unchanged between free actin, primed actomyosin and postPS actomyosin, except in the N-terminal residues, which are disordered in free actin, become ordered in primed actomyosin and adopt a different conformation in postPS actomyosin (Extended Data Fig. 7). It is important to note that the natural actin substrate of myosin-5 is cytoplasmic actin (β-actin and γ-actin), rather than the skeletal α-actin used here, and that these actins have N termini with slightly different sequences but conserved negative charge such that the interactions reported here are probably maintained. Density for the D-loop in actin subdomain 2 is also stronger in the actomyosin structures in comparison to free actin owing to stabilization following myosin binding.

When primed myosin binds to F-actin, actin residues 1–2 interact with both helix W and loop 2. The interactions with helix W provide stabilization of the L50 and cause a slight rotation of the U50 back towards the converter domain (Fig. 4b and Supplementary Video 2), which results in helix D movement creating strain in the nucleotide-binding pocket that would enable $P_i$ dissociation. However, $P_i$ cannot dissociate because the backdoor is still closed (Fig. 4e,f). These initial movements catalyse a subsequent rearrangement of the actin N-terminal residues that change their interactions with loop 2 and helix W, so that the C-terminal end

of loop 2 is stabilized (Fig. 3i) and actin E4 interacts with the activation loop. The stabilization of loop 2 at its C-terminal end means that the U50 domain and strut[26] are pulled towards the actin surface, promoting cleft closure (Fig. 4c). As interactions of the U50 with the actin surface are formed, committing myosin to cleft closure, switch 1 and the P-loop are moved away from switch 2, opening the backdoor, and concomitant reshaping of the nucleotide-binding pocket pushes $P_i$ into the $P_i$ release tunnel (Fig. 4g). Thus, actin catalyses myosin ATPase activity by accelerating cleft closure and $P_i$ dissociation.

Cleft closure is made energetically favourable in the presence of actin, owing to the formation of additional interfaces between the myosin U50 and −actin, and the distortions that occur following binding of primed myosin to actin act to accelerate $P_i$ release. Interactions of $P_i$ with positively charged residues in the $P_i$ release tunnel[20] could delay its release into solution and explain why kinetic[11,33] and single-molecule measurements[17] suggest that $P_i$ is released after the power stroke occurs[14]. Cleft closure causes the transducer to twist and the relay helix to straighten, concomitant with lever swing, producing the power stroke and the postPS structure (Fig. 4h).

In the absence of load, there is tight coupling between cleft closure and lever swing. However, under strain, the lead head of two-headed myosin-5 has been shown to rapidly release $P_i$ (ref. 34) yet adopt a strongly bound state with a primed lever[35,36]. This is consistent with $P_i$ displacement preceding cleft closure, which commits myosin to lever movement. The activation loop may also have a role here in stabilizing the actomyosin interface to decrease detachment under load[37].

The sequence of structures that we observe provides an additional insight into the way myosin works. The motions of cocking back around the actin axis and cleft closure are in planes almost orthogonal to that of lever swing, such that neither would be impeded in the presence of load on the lever along the F-actin axis. Thus, myosin clamps itself onto actin without producing any axial movement. Thus, when the lever tries to swing forwards against a restraining force, the axial force does not tend to reopen the cleft. This is akin to how a chameleon climbs up a stick. This feature has important implications for function across all myosin classes.

## Implications for two-headed myosin-5

By overlaying the structures of the primed and postPS actomyosin states, we were able to visualize the lever swing along actin (Fig. 5a–c). If we extend our structures to full lever length (Methods), the axial working stroke is about 34 nm, which is consistent with the distance between preferred binding sites on actin[7] (Fig. 5a). There is a small (4°) right-handed component to the lever swing (Fig. 5b,c) and a small (2.5°) right-handed torsion of the lever around its own axis, such that the lever tips are displaced from one another approximately 7° azimuthally around the actin axis.

To mimic the walking molecule, we placed a postPS and primed motor, with full-length levers, 13 actin subunits apart along an actin filament, as if they were the trailing and leading heads, respectively, of a myosin-5 double-headed molecule (Fig. 5d–f). The ends of the two levers were slightly displaced from each other azimuthally but were close axially along the filament. This shows that only slight bending of the levers or variation in actin helical symmetry[7] is needed to unite the heads onto the coiled-coil tail, as is observed by EM[35,38]. During walking, there is thus no need for a forward diffusive search by the detached head.

Together, these findings show that the myosin-5 motor is able to generate motion very effectively, producing an almost linear motion over a distance that is close to the typical step size along actin and is thus well adapted to its cargo-carrying function.

## Conclusions

By use of time-resolved cryo-EM, we have captured an actomyosin complex in the primed state and solved its structure to a global resolution

of 4.4 Å. Primed myosin initially binds actin through its lower 50-kDa subdomain. Owing to the high conservation in the primed actomyosin interface, the structure of this state is probably conserved across myosin classes and as such provides a valuable model for understanding the effects of disease-causing myosin mutations. Our time-resolved data show a primed actomyosin structure transitioning to a postPS structure, directly demonstrating the swinging lever mechanism, and enabling us to propose a mechanism for how actin catalyses both the structural transition and the release of the products of ATP hydrolysis.

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

# Methods

## Sample preparation

Rabbit skeletal actin in monomeric form (globular actin (G-actin)) was prepared as previously described[39]. Polymerization to F-actin was performed by mixing about 300 μM G-actin with 10% (v/v) cation exchange buffer (3 mM MgCl$_2$, 11 mM EGTA, pH 7.0), incubating for 5 min on ice, adding 10% (v/v) polymerization buffer (120 mM MOPS, 300 mM KCl, 12 mM MgCl$_2$, 1 mM EGTA, pH 7.0) and incubating the mixture overnight on ice. Mouse myosin-5a head fragment (subfragment 1, S1), coding for amino acids 1–797 (one IQ calmodulin-binding motif) and carrying the switch 1 S217A substitution, loop 2 DDEK(594–597) deletion and C-terminal Flag purification tag (Supplementary Fig. 2), was expressed using pVL1392 baculovirus transfer vector and purified as previously described[5]. Disodium ATP was obtained from Roche, and ADP was obtained from Sigma Aldrich.

## Kinetic measurements

Transient kinetics of actomyosin ATP hydrolysis were measured by use of a Hitech Scientific stopped-flow apparatus with single or double mixing, as appropriate. All stopped-flow experiments were carried out at 20 °C with a final buffer concentration of 37.5 mM potassium acetate, 25 mM KCl, 10 mM MOPS (pH 7.0), 2.25 mM MgCl$_2$, 0.1 mM EGTA, 0.25 mM dithiothreitol in the cell. See Supplementary Fig. 3 for specific method information.

## Time-resolved cryo-EM grid preparation

Time-resolved cryo-EM experiments were carried out using a custom-built set-up previously described[21] with modifications to allow two mixing steps. A photo and schematic of the set-up are shown in Supplementary Fig. 1. The flow rates for each individual syringe were 2.1 μl s$^{-1}$. In the first mixing step, myosin-5 at 51 μM in 10 mM MOPS, 100 mM KCl, 3 mM MgCl$_2$, 0.1 mM EGTA pH 7.0 was mixed 1:1 with 1 mM ATP in reaction buffer (10 mM MOPS, 50 mM potassium acetate, 2 mM MgCl$_2$, 0.1 mM EGTA pH 7.0). The mixture of myosin and nucleotide at a flow rate of 4.2 μl s$^{-1}$ was met by two 2.1 μl s$^{-1}$ flows of F-actin at 25 μM (subunit concentration in reaction buffer) in the flow-focusing region of the spray nozzle to create a mixture of actin and myosin comprising 13 μM myosin, 13 μM actin, 250 μM ATP, 10 mM MOPS, 38 mM potassium acetate, 25 mM KCl, 2 mM MgCl$_2$ and 0.1 mM EGTA at pH 7.0, and a total flow rate of 8.4 μl s$^{-1}$. This final mixture was sprayed onto an EM grid.

The average time delay from the first mixing step to the spray nozzle was 2.2 s, given a flow rate of 4.2 μl s$^{-1}$, tube length of 7 cm, inner diameter of 0.38 mm and dead volumes of 1.0 and 0.3 μl for mixer and nozzle, respectively. The spray nozzles used here have been described and characterized previously[4,40]. The nozzle-to-grid distance at the point of sample application was 1.3 cm, and the droplet speed was ≥30 m s$^{-1}$, resulting in a time-of-flight for the droplets of less than 1 ms. With a vertical distance of 1.7 cm between the spray nozzle and the liquid ethane surface and a grid speed of 1.8 m s$^{-1}$, the time delay was calculated to be 10 ms (10 ± 2 ms). The nozzle was operated in spraying mode with a spray gas pressure of 2 bar.

A longer time delay of about 120 ms was obtained by increasing the vertical distance between the nozzle and the ethane surface to 5.2 cm and pausing the grid after passing the spray. In these experiments, the sample mixture was incubated for an additional ≈100 ms on-grid, before plunging it into liquid ethane for vitrification. The total time delay from droplet application to vitrification was 120 ms (122 ± 5 ms), including deceleration, 100 ms pause and acceleration. Otherwise, the conditions for grid preparation were the same as for the 10 ms time point.

All grids were prepared at room temperature (about 20 °C) and at >60% relative humidity in the environmental chamber of the time-resolved EM device. Self-wicking grids were supplied by SPT Labtech and used after glow discharge in a Cressington 208 Carbon coater with a glow-discharge unit for 60 s at 0.1 mbar air pressure and 10 mA. Four replicate grids were prepared for each time point, three of which were taken forward for data collection.

A typical feature of grids made by spraying is thicker ice compared to those prepared using more standard approaches[41], such as by the use of the Vitrobot. This is due to the requirement for the drops to 'thin' after deposition on the grid, which can be limited by the short time between spraying and vitrification. This increase in ice thickness will be an important factor in limiting the resolution to about 4.0 Å, alongside conformational flexibility[18].

## Rigor actomyosin cryo-EM grid preparation

A 3 μl volume of pre-mixed F-actin and myosin (1:1 ratio with a final concentration of 1 μM) was applied to a Quantifoil R2/2 300 mesh grid that had been glow-discharged in a Cressington 208 Carbon coater for 30 s at 0.1 mbar air pressure and 10 mA before use. Grids were prepared by use of a Vitrobot Mark IV (ThermoFisher), with a blot time of 6 s and a blot force of 6 at 4 °C and 95% humidity with vitrification in liquid ethane.

## EM data gathering and processing

EM data were collected on a Titan Krios microscope equipped with a Gatan K2 direct electron detector operated in counting mode using the EPU software version 2.12 distributed by Thermo Scientific. The main data collection and processing parameters for the time-resolved EM grids are listed in Extended Data Table 1. A schematic overview of the processing pipeline is given in Supplementary Fig. 4. Data from three grids were collected for each time point. All processing was carried out using RELION-3.1 (ref. 42), unless otherwise mentioned. Micrographs were corrected for beam-induced motion using MotionCor2, and CTF estimation was carried out using GCTF[43,44]. Actin filaments were manually picked and processed using standard helical processing methods[45] (Supplementary Fig. 4). After CTF-refinement and Bayesian polishing, all six datasets were combined, and a helical consensus structure was calculated (Supplementary Fig. 4c). Using focused three-dimensional (3D) classification without alignment (non-helical) and a mask that covered the central myosin-binding site (Supplementary Fig. 4d), particles were classified into actomyosin states or bare actin, with a small fraction of particles left unassigned. Despite performing multiple rounds of 3D classification, we found that actomyosin particles were classified into either a primed actomyosin state, with a primed lever and an open actin-binding cleft, or a postPS actomyosin state, with a postPS lever and a closed actin-binding cleft (Supplementary Fig. 4d,e). No other actomyosin states were identified. The final reconstruction of free actin was obtained by helical refinement. Primed and postPS actomyosin were refined helically and after partial signal subtraction, as single particles (Supplementary Fig. 4c–e). Post-processing was performed in RELION and in DeepEMhancer[46].

To quantify the conversion of primed actomyosin to postPS actomyosin between the 10 ms and 120 ms time points, and consistency of this between grids, each particle used in the primed and postPS structures was assigned back to the grid from which it was imaged, and then the total number of primed particles was expressed as a proportion of all particles assigned to each grid.

For unbound myosin-5, the processing parameters are listed in Extended Data Table 1, with an overview of the processing pipeline shown in Supplementary Fig. 5. Unbound myosin-5 particles were picked from a subset of micrographs of the 120 ms time-resolved data. As a result of thicker ice, unbound myosin particles were not picked from the 10-ms data. After one round of 2D classification, good particles were used to train a crYOLO model[47]. With the trained model, particles were picked from the entire 120-ms dataset, leading to a final selection of 23,930 particles after 1 round of 2D and 1 round of 3D classification. The final 3D refinement after Bayesian polishing was performed using non-uniform refinement in cryoSPARC[48].

For rigor actomyosin-5, the processing parameters are listed in Extended Data Table 3, with an overview of the processing pipeline shown in Supplementary Fig. 6. Particles were manually picked from a subset of micrographs, and after one round of 2D classification, good particles were used to train a crYOLO model[47]. With the trained model, particles were picked from the entire rigor dataset, leading to a final selection of 85,986 particles after 1 round of 2D classification. The particles then underwent two rounds of 3D helical refinement and Bayesian polishing. Following particle polishing, signal outside the masked actomyosin complex (myosin and the central three actin subunits) was subtracted. The subtracted particles then underwent 2 rounds of focused classification on the myosin motor domain leading to a final selection of 21,890 particles. The final 3D refinement was then performed using non-uniform refinement in cryoSPARC 3.2.0 (ref. 48).

## Pseudo-atomic models

Homology pseudo-atomic models for primed and postPS actomyosin-5 structures were generated using Modeller in Chimera on the basis of the Protein Data Bank (PDB) files shown in Extended Data Table 1 (refs. 49,50). Each model comprised three actin subunits and one myosin-5 motor domain. Refinement of these models was performed using Coot[51], with subsequent refinement of the nucleotide pocket in ISOLDE, implementing the hydrogen bonding coordination to the $P_i$ groups as described in ref. 5 and Y119 coordination as described in ref. 18 as harmonic restraints during flexible fitting[28]. Real-space refinement was performed using Phenix[52].

To permit elucidation of interactions occurring at the actomyosin interface, we used MD simulations as a further model refinement tool. These were performed with the Amber FF14SB forcefield and a GBSA (generalized Born with solvent-accessible surface area) implicit solvent model following the method described in ref. 22 with position restraints on all backbone atoms using a restraint weight of 10 kcal mol$^{-1}$ Å$^{-2}$ and a production run time of 3 ns. Interactions that were observed for at least half of the simulation time were included in the pseudo-atomic model. The D-helix and the actin N-terminal interactions with myosin were further refined by use of ISOLDE[28] using default parameters and implementing the hydrogen bonding coordination to the nucleotide as described in refs. 5,18 as harmonic restraints during flexible fitting of the D-helix.

To enable interpretation of our mutant rigor actomyosin-5 cryo-EM density map, we rigidly fitted existing rigor actomyosin-5 PDB structures generated by ref. 18, into the map. The myosin chain from PDB ID 7PLV was well accommodated into the map, especially within the converter region (Extended Data Fig. 6f). Therefore, we used this as a model for our rigor myosin-5 structure to enable us to compare it to our postPS actomyosin-5 structure (Extended Data Fig. 6f,g).

To illustrate our proposed mechanism of force generation and actin activation (Supplementary Video 4), we separated out the motions of cleft closure and power stroke into a suggested time sequence. To achieve this, a chimeric model of primed and postPS actomyosin was generated to represent a midway point in the primed-to-postPS transition of myosin. This included myosin chain numbering: amino acids 1–128 primed, amino acids 129–449 postPS, amino acids 450–507 primed, amino acids 508–632 postPS, amino acids 633–763 primed. Structures were visualized in Chimera. Videos were generated by use of Chimera, Adobe Aftereffects and Adobe Premiere.

## MD simulations

We originally used implicit solvent MD simulations, with short (3 ns) production run times, to establish interactions that may be occurring at the actomyosin interface. To test whether the side-chain interactions observed in our primed and postPS actomyosin models at the interface between actin and loop 2 and between actin and helix W persist over longer times when subjected to thermal fluctuations, we performed longer MD simulations (>100 ns; wall clock time of about 5 days), with three independent replicates in explicit solvent conditions for each model. All simulations were run in AMBER16 (refs. 53,54; GPU version PMEMD-CUDA) on all atom systems parameterized with the charmm36m forcefield[55] and built using CHARMM-GUI[56]. The primed and postPS actomyosin structures were prepared for MD simulations using the CHARMM-GUI solution builder[56–58]. The ADP present in both structures and the $P_i$ present in the primed structure were parameterized from the CHARMM-GUI library with the $P_i$ built as $H_2PO_4$. All N termini present in the models were acetylated and C termini were made neutrally charged. A TIP3P octahedral water box (Supplementary Fig. 7) was built for each structure using 25 mM KCl, 38 mM potassium acetate and 2 mM $MgCl_2$ (pH 7.0) with a diameter of 186 Å for primed and 172 Å for postPS actomyosin; each system had a total atom count of 468,219 and 369,572 for primed and postPS, respectively. The simulations were then run at a constant temperature of 300 K and constant pressure ensemble (NPT). Simulations were run with position restraints on all backbone atoms using a restraint weight of 5 kcal mol$^{-1}$ Å$^{-2}$, allowing side chains to move. Each condition was run in triplicate using a different random seed to initiate the starting velocities. Trajectories (Supplementary Videos 5 and 6) were analysed using Chimera, and distance plots for atom–atom interaction distances were prepared using GraphPad Prism (Extended Data Fig. 2). The percentage of time that interacting atoms were within H-bonding distance (3.3 Å) was outputted (Extended Data Table 2).

## Myosin-5 full-length lever model

A model F-actin filament 17 subunits long was created by 7 superpositions of our 3-actin-subunit model. Full-length levers (to residue 909) were added onto our primed and postPS actomyosin structures by super-imposing levers from PDB ID 7YV9 chain A, aligned on the converter domain (residues 699–750). Lever swing and azimuthal displacement were measured using the measurement tools in Chimera.

## Reporting summary

Further information on research design is available in the Nature Portfolio Reporting Summary linked to this article.

## Data availability

The electron density maps and atomic models for unbound primed myosin-5, primed actomyosin-5 and postPS actomyosin-5 have been deposited into the Electron Microscopy Data Bank (EMDB), with the accession codes EMD-19031, EMD-19013 and EMD-19030, and the PDB with the accession codes 8RBG, 8R9V and 8RBF, respectively. The electron density map for rigor actomyosin-5 has been deposited into the EMDB with the accession code EMD-50594. The following models were used for comparison purposes in our study—actomyosin-5 rigor structures PDB IDs: 7PLT, 7PLU, 7PLV, 7PLW and 7PLZ; and actomyosin-5 strong-ADP structures PDB IDs: 7PM5, 7PM6, 7PM7, 7PM8 and 7PM9. Each MD trajectory and parameter–topology file for primed and postPS structures for each MD simulation replicate is provided along with the starting models at figshare via https://doi.org/10.6084/m9.figshare.24948180 (ref. 59).

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

**Acknowledgements** H. White, and the late J. Trinick, proposed this experiment some 40 years ago, and it has taken until now, with the required improvements in technology, for it to be accomplished. We thank J. Trinick for foresight and mentorship; members of the cryo-EM community at Leeds for help and guidance, particularly G. Carrington for advice on video generation; and the support scientists at the Astbury Biostructure Laboratory for help in EM data acquisition. All EM data were collected at the Astbury Biostructure Laboratory financed by the University of Leeds and the Wellcome Trust (108466/Z/15/Z). MD simulations were undertaken on ARC4, part of the High Performance Computing facilities at the University of Leeds, UK. This work was financed by a Biotechnology and Biological Sciences Research Council grant to S.P.M. (BB/P026397/1) and supported by research grants to H.D.W. from the American Heart Association, to H.D.W. and V.E.G. from the US National Institutes of Health (NIHR21AR-071675), and to S.A.H. from the UK Engineering and Physical Sciences Research Council (EP/T026308/1 and EP/T026308/2). This work was also supported by equipment purchased by a Royal Society grant to C.A.S. (RGS\R1\231276). S.N.M. is supported by a PhD studentship financed by the School of Molecular and Cellular Biology, University of Leeds. D.P.K. was a PhD student on the Wellcome Trust four-year PhD programme in The Astbury Centre funded by The University of Leeds. C.A.S. is supported by a British Heart Foundation Jacqueline Murray Coomber Fellowship (FS/20/21/34704).

**Author contributions** H.D.W. and S.P.M. designed the project. B.V. and E.F. produced the myosin mutant constructs. H.D.W., S.P.M., D.P.K. and F.S. aided in design of the time-resolved approach. D.P.K., S.P.M. and H.D.W. performed time-resolved cryo-EM grid screening, optimization and data collection. D.P.K., C.R. and V.E.G. performed initial data analysis and initial processing of the cryo-EM data. E.F., J.L.A., H.D.W. and D.A.W. performed kinetic experiments and kinetic data analysis. M.S. performed kinetic modelling. S.N.M. and C.A.S. performed cryo-EM data refinement and final model building. S.N.M. and S.A.H performed MD simulations. S.N.M. and C.A.S. performed structure validation. S.N.M., C.A.S., S.P.M., P.J.K., S.A.H., D.P.K. and H.D.W. interpreted the data and the model. S.N.M and C.A.S. performed main figure and video generation. S.N.M., C.A.S., D.P.K., J.L.A., and S.P.M. produced supplementary figures. C.A.S., S.P.M., P.J.K., D.P.K., S.N.M, and H.D.W. wrote the manuscript. All authors discussed the results and commented on the manuscript.

**Competing interests** The authors declare no competing interests.

**Additional information**
**Correspondence and requests for materials** should be addressed to Stephen P. Muench, Charlotte A. Scarff or Howard D. White.

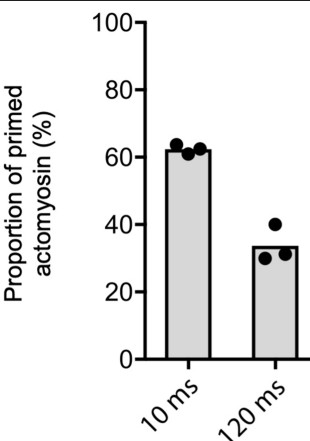

**Extended Data Fig. 1 | Time dependence of the proportion of primed state actomyosin.** Proportion of the primed state vitrified at 10 ms or 120 ms after mixing myosin and F-actin. Shown is the mean as grey bars and replicates as black points, n = 3 biologically independent (discrete) experiments for the 10 ms and 120 ms time points respectively.

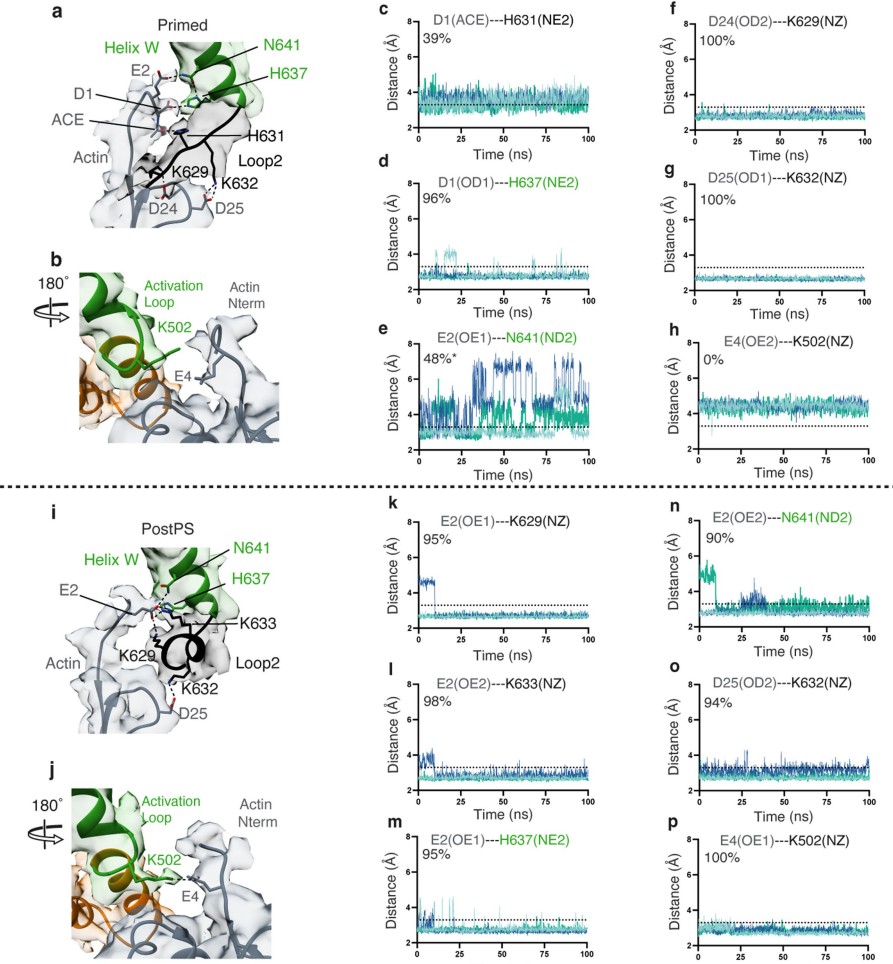

**Extended Data Fig. 2 | Molecular dynamics analysis of actin N-terminal interactions with myosin loop2, helixW and the activation loop.**
**a**,**b**, Portion of the primed actomyosin density map and fitted structures, displaying the putative interactions between -actin's N-terminal region and myosin (**a**) helixW/loop2 and (**b**) activation loop. **c**–**h**, Time courses of the distance between atoms involved in interactions in primed actomyosin during three 100 ns molecular dynamics simulations (cyan, sea green, blue). Each interaction is identified at the top of its panel. ACE refers to the oxygen atom of -actin D1. **i**,**j**, Same as **a**,**b** but showing the equivalent views for the postPS structure. **k**–**p**, Same as **c**–**h** but showing time courses of putative

interactions in the postPS structure. For all time courses, the average % time that the interatomic distance is within bonding range (3.3 Å; horizontal dotted line in each time course) across the three replicates is shown in the top left of each plot, and within Extended Data Table 2. Note that the most significant change in distance between atoms involved in interactions is for E2(OE1)-N641(ND2) shown in panel **e**. This is because, within the simulations, E2 switched between interacting with N641(ND2) via its OE1 and its OE2 atom. N641 also switches side chain position to form metastable interactions with H637. Despite these alternative interactions, the interaction shown is populated the most (48 % of the time).

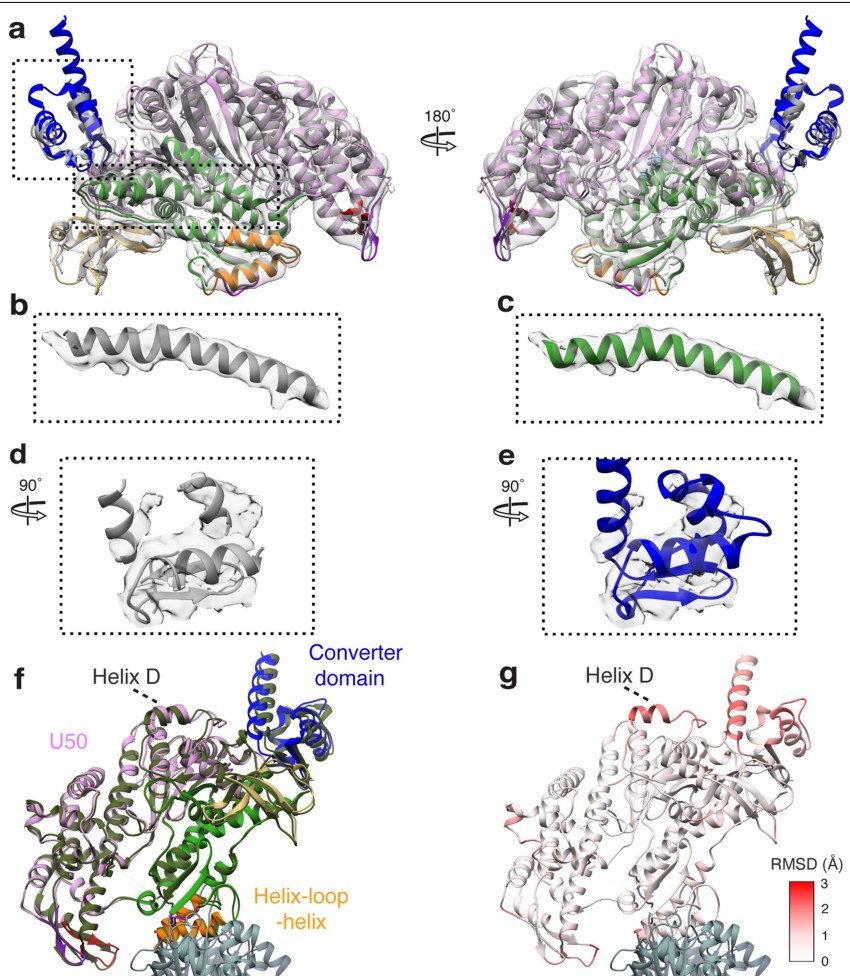

**Extended Data Fig. 3 | Atomic model of myosin-5a in the primed state. a**, The EM density map of unbound myosin-5 with the crystal structure (PDB ID 4zg4) fitted directly (grey) and after flexible fitting (with subdomains colored, U50 pink, L50 green, N-term gold, converter blue, HLH orange). **b**, Relay helix from PDB ID 4zg4 fitted into EM density map for unbound myosin-5 directly and **c**, after flexible fitting. **d**, Converter domain of PDB ID 4zg4 fitted into EM density map for unbound myosin-5 directly and **e**, after flexible fitting. **f**, Global superposition of the primed actomyosin-5 (coloured as in Fig. 1) and unbound primed myosin-5 (coloured olive green) shows a similar structure with no significant changes in domain architecture. **g**, r.m.s.d. of myosin residues between primed actomyosin and primed myosin.

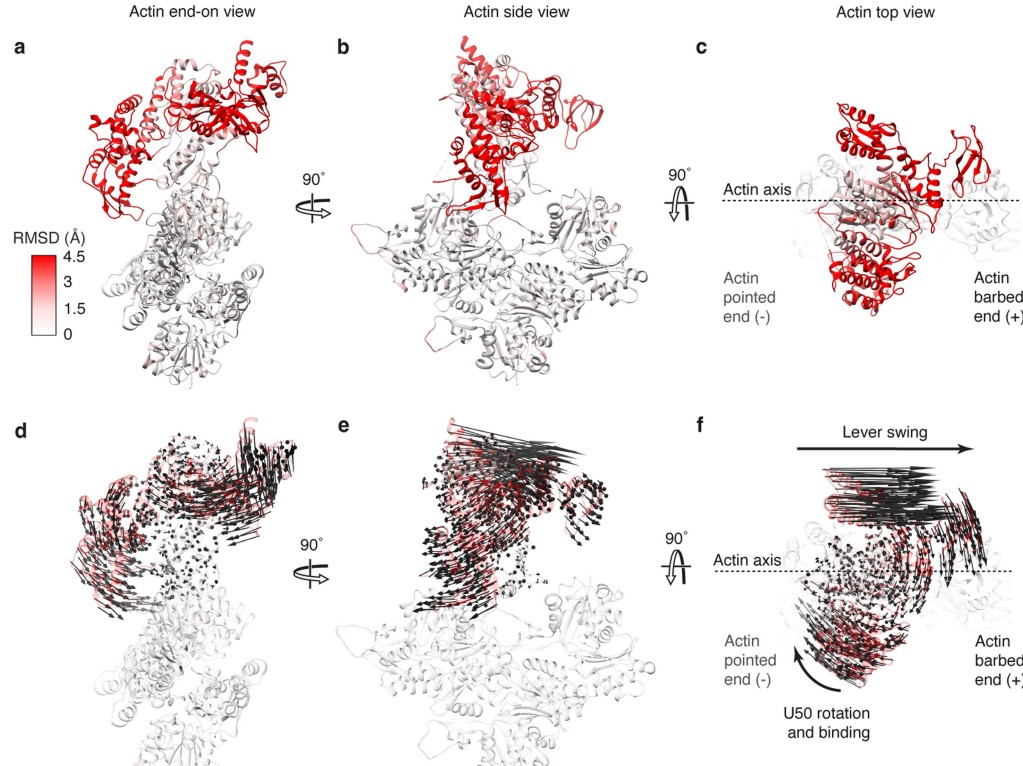

**Extended Data Fig. 4 | RMSD between primed and postPS actomyosin structures. a–c**, Primed actomyosin PDB coloured by RMSD between primed and postPS actomyosin shown in **a**, end-on view of F-actin, looking towards the pointed end with myosin bound to the top of the actin; **b**, side view; **c**, top view, looking down over the motor domain. **d–f**, The same views as **a–c** respectively, but with vector arrows (in black) showing displacement in relative Cα positions between primed and postPS actomyosin motor domains.

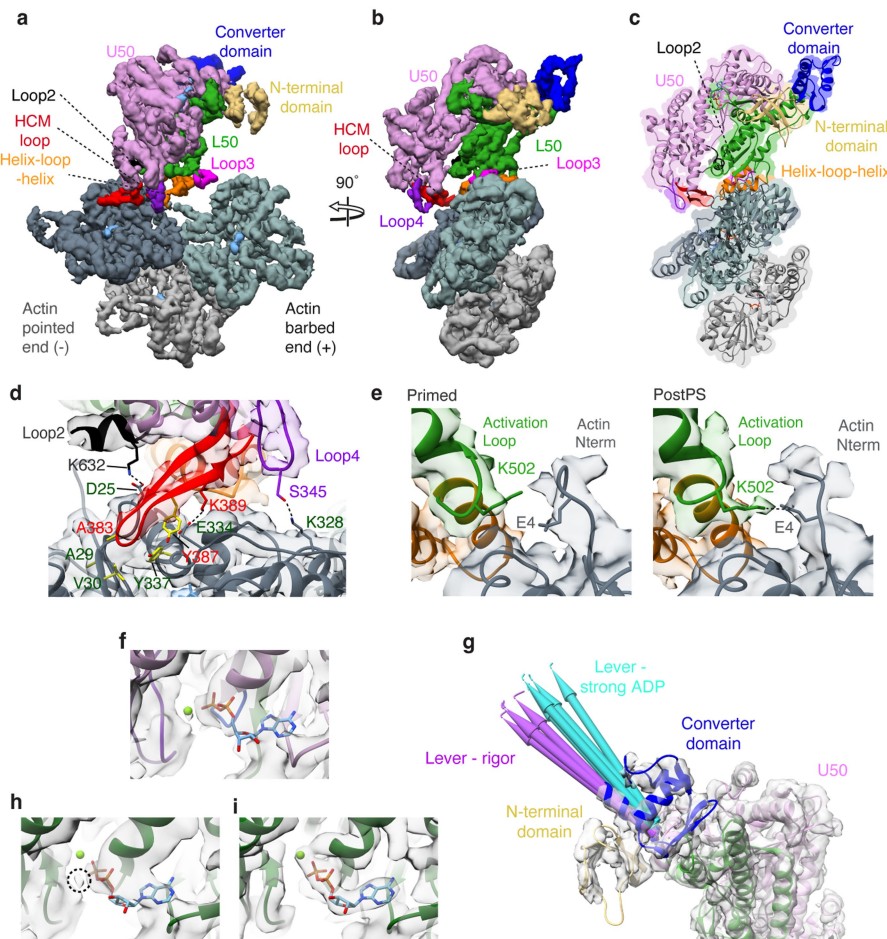

**Extended Data Fig. 5 | PostPS actomyosin structure.** CryoEM density map of the postPS actomyosin-5 complex, segmented and coloured by myosin subdomains and actin chains as indicated (with central three actin subunits displayed). Map thresholded to show secondary structure (threshold 0.15). Shown in (**a**) side view of actin and in (**b**) in end-on view of F-actin, looking toward the pointed end. **c**, Backbone depiction of atomic model of postPS actomyosin-5, fitted into the EM density map, with view as in **b**. Actin subunits are shown in slate grey (−end), blue-grey (+end), and light grey. **d**, Magnified view of the U50, loop2, HCM loop and loop4 contacts to actin. Relevant interacting residues are labelled and shown with hydrophobic residues in yellow. **e**, Magnified view of the primed and postPS actomyosin states focused on the activation loop-actin N-terminal interaction interface to show formation of a salt bridge between K502 and E4 only in the postPS structure. **f**, PostPS nucleotide pocket fit to EM density (map threshold 0.0096). **g**, PostPS structure, focused on converter domain. The lever position is more consistent with that observed in previous actomyosin-5 rigor structures (purple pipes, PDB IDs: 7PLT, 7PLU, 7PLV, 7PLW, 7PLZ) than actomyosin-5 strong-ADP structures (turquoise pipes, PDB IDs: 7PM5, 7PM6, 7PM7, 7PM8, 7PM9). **h**, Nucleotide pocket of actomyosin structure 7PM5 fitted to our postPS EM density highlighting unfilled magnesium density with a dashed circle. **i**, Nucleotide pocket of actomyosin structure 7MP5 fitted to corresponding density EMDB ID: 13521 (map threshold 0.0197). DeepEMhancer post-processed map depicted in **a**–**d**, **g**, and *RELION post-processed map in **e**,**f**,**h**,**i**.

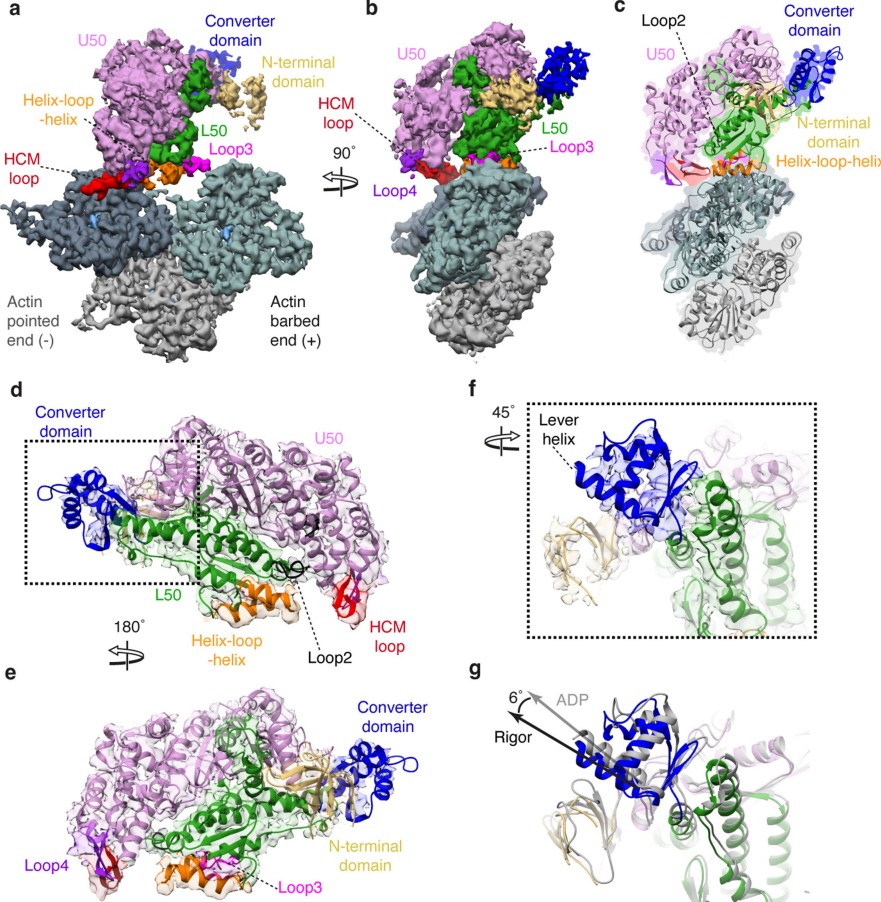

**Extended Data Fig. 6 | Rigor actomyosin structure.** CryoEM density map of our rigor actomyosin-5 complex, segmented and coloured by myosin subdomains and actin chains as indicated (with central three actin subunits displayed). Map thresholded to show secondary structure (threshold 0.014). Shown in (**a**) side view of F-actin and (**b**) end-on view of F-actin, looking toward the pointed end. **c**, Backbone depiction of atomic model of rigor myosin-5 structure PDB ID 7PLV and actin from our postPS actomyosin-5 structure, rigidly fitted into the EM density map, with view as in **b**. Actin subunits are shown in slate grey (−end), blue-grey (+end), and light grey. **d**,**e**, Focused views of myosin-5 7PLV model in our rigor cryoEM density map segmented and coloured by myosin subdomains as in **c** (threshold 0.014). **f**, Magnified view of the rigor 7PLV converter domain fit to rigor cryoEM density map. **g**, Comparison of lever conformation between 7PLV rigor coloured as in **d** and postPS coloured grey, highlighting the 6° displacement of lever helix on ADP release (aligned on actin).

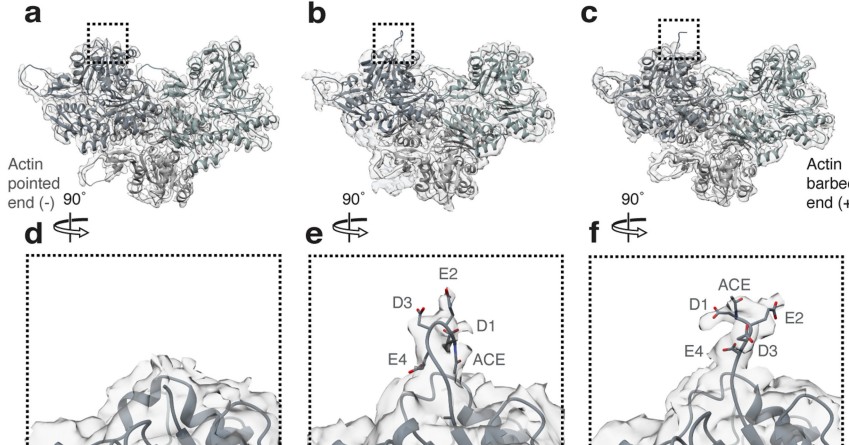

**Extended Data Fig. 7 | Actin structure.** Actin structure is preserved between (**a**) actin alone, (**b**) primed actomyosin, and (**c**) postPS actomyosin, except at the N-terminus where it becomes ordered when myosin binds. D-loop density also becomes more ordered when associated with myosin. The density observed for the N-terminal four residues of actin is absent in (**d**) vacant actin, and different between (**e**) primed actomyosin, and (**f**) postPS actomyosin. ACE denotes the acetyl group of D1.

**Extended Data Table 1 | Data collection, processing, and refinement statistics for time-resolved EM data, for three collections at 10 ms ([a,b,c]) and 120 ms ([e,f,g]) respectively**

| | 10ms | 120ms |
|---|---|---|
| **Data collection and processing** | | |
| Magnification | 130,000 x | 130,000 x |
| Voltage (kV) | 300 | 300 |
| Electron exposure (e–/Å$^2$) | 67.58[a] | 55.4[e] |
| | 55.4[b] | 56.5[f] |
| | 61.2[c] | 56.5[g] |
| Defocus range (μm) | -2 to -4.1 | -2 to -4.1 |
| Pixel size (Å) | 1.07 | 1.07 |

| | Primed actomyosin | PostPS actomyosin | Unbound myosin |
|---|---|---|---|
| | (EMDB-19013) (PDB 8R9V) | (EMDB-19030) (PDB 8RBF) | (EMDB-19031) (PDB 8RBG) |
| Initial particle images | 218,602[a] | 122,312[e] | 83,466[e] |
| | 285,704[b] | 122,312[f] | 560,987[f] |
| | 211,425[c] | 111,047[g] | 85,233[g] |
| Final particle images | 93374 | 94093 | 23930 |
| Resolution (Å) FSC (0.143) | 4.4 | 4.2 | 4.9 |
| **Refinement** | | | |
| Initial model used | 4ZG4 (myosin-5c), 5ONV (F-actin) | PDB 1W7I (myosin-5a), 5ONV (F-actin) | 4ZG4 (myosin) |
| Map sharpening *B* factor (Å$^2$) | -119 | -84 | -233 |
| Model composition | | | |
| Non-hydrogen atoms | 14852 | 14790 | 6008 |
| Protein residues | 1857 | 1848 | 737 |
| Ligands | 4 | 4 | 1 |
| R.M.S.Z deviations | | | |
| Bond lengths (Å) | 0.61 | 0.6 | 0.66 |
| Bond angles (°) | 0.75 | 0.77 | 1.04 |
| Validation | | | |
| MolProbity score | 2.06 | 1.75 | 1.70 |
| Clashscore | 17.12 | 9.73 | 10.03 |
| Poor rotamers (%) | 0.25 | 0.19 | 0.62 |
| Ramachandran plot | | | |
| Favored (%) | 96.34 | 96.41 | 97.00 |
| Allowed (%) | 4.34 | 3.54 | 2.73 |
| Disallowed (%) | 0.25 | 0.05 | 0.27 |

Data from all six datasets were combined and processed to give rise to the primed and postPS actomyosin cryoEM density maps. Data from the 120 ms dataset were used to produce the unbound myosin-5 cryoEM density map.

**Extended Data Table 2 | Actin N-terminal interactions with myosin-5, derived from molecular dynamics analysis with each interacting residue and atom specified**

| Primed actomyosin | | |
|---|---|---|
| Actin residue (atom ID) | Myosin residue (atom ID) | % time of interaction |
| D1 (O-ACE) | H631 (NE2) | 39 |
| D1 (OD1) | H637 (NE2) | 96 |
| E2 (OE1) | N641 (ND2) | 48* |
| D24 (OD2) | K629 (NZ) | 100 |
| D25 (OD1) | K632 (NZ) | 100 |
| E4 (OE2) | K502 (NZ) | 0 |
| PostPS actomyosin | | |
| Actin residue (atom ID) | Myosin residue (atom ID) | % time of interaction |
| E2 (OE1) | K629 (NZ) | 95 |
| E2 (OE2) | K633 (NZ) | 98 |
| E2 (OE1) | H637 (NZ2) | 95 |
| E2 (OE2) | N641 (ND2) | 90 |
| D25 (OD2) | K632 (NZ) | 94 |
| E4 (OE1) | K502 (NZ) | 100 |

The % time of interaction was determined by calculating the percentage time the atoms specified were within a 3.3 Å distance, across three independent simulations of >100 ns each. O-ACE refers to the oxygen atom of the acetyl moiety of D1. *N641 (ND2) was observed to interact with both the OE1 and OE2 oxygens of actin residue E2 and the % interaction time was 51% when averaged across both E2 oxygens.

**Extended Data Table 3 | Data collection and processing parameters for rigor actomyosin data**

| | Rigor actomyosin (EMDB-50594) |
|---|---|
| **Data collection and processing** | |
| Magnification | 130,000 x |
| Voltage (kV) | 300 |
| Electron exposure (e–/Å$^2$) | 56.5 |
| Defocus range (μm) | 1 to -3 |
| Pixel size (Å) | 1.07 |
| Initial particle images (no.) | 128932 |
| Final particle images (no.) | 21890 |
| Map resolution (Å) FSC (0.143) | 4.9 |

# Reporting Summary

## Statistics

For all statistical analyses, confirm that the following items are present in the figure legend, table legend, main text, or Methods section.

| n/a | Confirmed | |
|---|---|---|
| ☐ | ☒ | The exact sample size ($n$) for each experimental group/condition, given as a discrete number and unit of measurement |
| ☐ | ☒ | A statement on whether measurements were taken from distinct samples or whether the same sample was measured repeatedly |
| ☒ | ☐ | The statistical test(s) used AND whether they are one- or two-sided<br>*Only common tests should be described solely by name; describe more complex techniques in the Methods section.* |
| ☒ | ☐ | A description of all covariates tested |
| ☒ | ☐ | A description of any assumptions or corrections, such as tests of normality and adjustment for multiple comparisons |
| ☐ | ☒ | A full description of the statistical parameters including central tendency (e.g. means) or other basic estimates (e.g. regression coefficient) AND variation (e.g. standard deviation) or associated estimates of uncertainty (e.g. confidence intervals) |
| ☒ | ☐ | For null hypothesis testing, the test statistic (e.g. $F$, $t$, $r$) with confidence intervals, effect sizes, degrees of freedom and $P$ value noted<br>*Give P values as exact values whenever suitable.* |
| ☐ | ☒ | For Bayesian analysis, information on the choice of priors and Markov chain Monte Carlo settings |
| ☒ | ☐ | For hierarchical and complex designs, identification of the appropriate level for tests and full reporting of outcomes |
| ☒ | ☐ | Estimates of effect sizes (e.g. Cohen's $d$, Pearson's $r$), indicating how they were calculated |

*Our web collection on statistics for biologists contains articles on many of the points above.*

## Software and code

Policy information about availability of computer code

Data collection
Data were collected on a Titan Krios microscope equipped with a Gatan K2 direct electron detector operated in counting mode using the EPU software version 2.12 distributed by Thermo Scientific.

Data analysis
Data processing was done using RELION-3.1 and cryoSPARC version 3.2.0. Micrographs were corrected for beam-induced motion using MotionCor2 and CTF estimation was done using GCTF. Post-processing was performed in RELION and in DeepEMhancer. Homology models were generated using Modeller within Chimera. Model building was done using coot, with subsequent refinement of the nucleotide pocket in ISOLDE. Real space refinement was performed using Phenix. Videos were generated by use of Chimera, Adobe Aftereffects and Adobe Premiere. All simulations were run in AMBER16 (GPU version PMEMD-CUDA) on all atom systems parameterised with the charmm36m forcefield and built using CHARMM-GUI. The primed and postPS actomyosin structures were prepared for molecular dynamics simulations using the CHARMM-GUI solution builder.

For manuscripts utilizing custom algorithms or software that are central to the research but not yet described in published literature, software must be made available to editors and reviewers. We strongly encourage code deposition in a community repository (e.g. GitHub). See the Nature Portfolio guidelines for submitting code & software for further information.

## Data

Policy information about availability of data

 All manuscripts must include a data availability statement. This statement should provide the following information, where applicable:

- Accession codes, unique identifiers, or web links for publicly available datasets
- A description of any restrictions on data availability
- For clinical datasets or third party data, please ensure that the statement adheres to our policy

The electron density maps and atomic models for unbound primed myosin-5, primed actomyosin-5 and postPS actomyosin-5 have been deposited into EMDB, with accession codes EMD-19031, EMD-19013 and EMD-19030, and the PDB with accession codes 8RBG, 8R9V and 8RBF, respectively. The electron density map for rigor actomyosin-5 has been deposited into the EMDB with accession code EMD-50594.
The following models were used for comparison purposes in our study, actomyosin-5 rigor structures PDB IDs: 7PLT, 7PLU, 7PLV, 7PLW, 7PLZ and actomyosin-5 strong-ADP structures PDB IDs: 7PM5, 7PM6, 7PM7, 7PM8, 7PM9.
Each MD trajectory and parameter-topology file for primed and PostPS structures for each MD simulation replicate is provided along with the starting models at: https://doi.org/10.6084/m9.figshare.24948180

## Research involving human participants, their data, or biological material

Policy information about studies with human participants or human data. See also policy information about sex, gender (identity/presentation), and sexual orientation and race, ethnicity and racism.

| | |
|---|---|
| Reporting on sex and gender | N/A |
| Reporting on race, ethnicity, or other socially relevant groupings | N/A |
| Population characteristics | N/A |
| Recruitment | N/A |
| Ethics oversight | N/A |

Note that full information on the approval of the study protocol must also be provided in the manuscript.

# Field-specific reporting

Please select the one below that is the best fit for your research. If you are not sure, read the appropriate sections before making your selection.

☒ Life sciences  ☐ Behavioural & social sciences  ☐ Ecological, evolutionary & environmental sciences

For a reference copy of the document with all sections, see nature.com/documents/nr-reporting-summary-flat.pdf

# Life sciences study design

All studies must disclose on these points even when the disclosure is negative.

| | |
|---|---|
| Sample size | Sample size was determined based on the required particle number to achieve the desired resolution |
| Data exclusions | Data were excluded during the 2D and 3D classification stage based on typical data processing procedures as shown in image processing figures |
| Replication | Time points (10 ms and 120 ms respectively) were repeated 3 times as described in the manuscript to show reproducibility |
| Randomization | Data processing approaches used (maximum likelihood classification) ensure randomization and unbiased particle classification |
| Blinding | Blinding was not required. Whilst different datasets were acquired at time points of 10 and 120 ms respectively these were combined and analyzed by the same methods. |

# Reporting for specific materials, systems and methods

We require information from authors about some types of materials, experimental systems and methods used in many studies. Here, indicate whether each material, system or method listed is relevant to your study. If you are not sure if a list item applies to your research, read the appropriate section before selecting a response.

## Materials & experimental systems

| n/a | Involved in the study |
|---|---|
| ☒ ☐ | Antibodies |
| ☒ ☐ | Eukaryotic cell lines |
| ☒ ☐ | Palaeontology and archaeology |
| ☒ ☐ | Animals and other organisms |
| ☒ ☐ | Clinical data |
| ☒ ☐ | Dual use research of concern |
| ☒ ☐ | Plants |

## Methods

| n/a | Involved in the study |
|---|---|
| ☒ ☐ | ChIP-seq |
| ☒ ☐ | Flow cytometry |
| ☒ ☐ | MRI-based neuroimaging |

## Plants

| Seed stocks | N/A |
|---|---|
| Novel plant genotypes | N/A |
| Authentication | N/A |

