## [Peer Review file · Nature]

Swinging lever mechanism of myosin directly shown by time-resolved cryoEM

Corresponding Author: Professor Stephen Muench

Version 0:

Reviewer comments:

Referee #1

(Remarks to the Author)

The manuscript by Kelbl et al presents a time-resolved cryo-EM study of the interaction of F-actin with myosin-V and its swinging lever mechanism. This is a fundamental biological process with very wide relevance to basic and translational science and it has been studied in great detail over many decades, in particular also by molecular biology, biophysical and structural means. The authors correctly point out the lack of insights into short-lived, transient on-pathway structural intermediates and thus attempt to address this by rapid mixing and snap-freezing of mixtures of a (slow hydrolysis) mutant form of primed myosin-V and F-actin at 10 and 120 ms time points. From a purely engineering perspective, this is no small achievement. However, the obtained biological insights are very limited. The authors were able to determine only two structures at resolutions just high enough to warrant tentative molecular interpretation. One of these represents a post power stroke actomyosin-V conformation, as is admitted by the authors, basically reproduces one of the dozens trapped structures of the pathway reported by the Raunser department (Pospich et al., eLife, 2021). In effect, this work reports a single novel structure, which falls far short of the claimed elucidation of the power stroke mechanism. As such, this manuscript would be of interest to researchers specialized in the structural biology of myosin and might be suitable for a specialized structural journal in a short report form. Even so, there are quite some major points that the authors would need to work on before it would be acceptable for publication. Thus, should a direct transfer to NSMB be of interest, careful attention should be paid to the below major points:

1. The claims in the manuscript are very inflated. At global resolutions of 4.4 and 4.9 Å and notably lower local resolution especially in functionally relevant loop regions, most of the modeled side chains and bonds are to be taken with the utmost caution. More importantly, the lack of any intermediate structures from primed actomyosin state to the post power stroke conformation, which is not novel but rather reproduces a conformation reported in 2021, should effectively force the authors to discuss their two structures in the context of the wealth of other structural work. Surprisingly, the authors instead present a very detailed molecular descriptions of molecular interfaces (including side chains in loops and salt bridges in areas of very low local resolution) and conformational rearrangements by 2-3 Å (i.e. smaller than the global resolution of the reported maps). Such notes, at best, belong in an extended supplementary text's discussion and figures and should also include equally detailed comparison to published work.
2. The authors worked with a double mutant of myosin-V which slows down hydrolysis. Even though it is understandable that they had to resort to such mutants, because the studied reaction would be faster than their technically possible time resolution, it is not clear from the description in the paper whether these mutants alter the molecular pathway. The authors would need substantial backing from orthogonal biophysical assays to ascertain that any trapped intermediates (i.e. presently the single one) is consistent with the wild type pathway.
3. The paragraph on "structural mechanism of myosin force generation and ATPase activation on F-actin" is not appropriate in the results part as it is based on a single novel structure. It is a well-written and important piece of text but much rather suitable for a review article.

In summary, this manuscript attempts a technically difficult time-resolved structural study of an important pathway but falls short of delivering any novel biological insights and presents very modest experimental results relative to the claims made. In order for the manuscript to be accepted in a peer-reviewed journal, it would be important to improve the resolution to a point that allows the authors to interpret the structure(s) at near-atomic resolution, in particular in the regions that appear flexible at the 10 ms time point as well characterize the mutant they used. Moreover, for the work to be of interest to a broad readership of a specialized and highly respected journal like NSMB or Nat Comms, it would be important to attempt to solve

further previously unknown states, comment what might be in the way of observing most of the intermediates, and thoroughly contextualise the results with respect to the wealth of previously reported structural intermediates.

Referee #2

(Remarks to the Author)

The manuscript "Swinging lever mechanism of myosin demonstrated by time-resolved cryoEM" by Klebl et al uses time resolved cryoEM and a myosin 5 mutant with slowed Pi release and increased actin affinity to capture a 4.4Å resolution structure of the primed actomyosin complex, which has previously not been visualized. The approach is novel, the experiments well conceived, and the manuscript well written. I believe it is of sufficiently broad interest to warrant publication in Nature with the following issues addressed.

1. Claims involving side chain interactions are not warranted – the resolution of the map does not allow one to see individual side chains. This needs to be recognized when reporting, for example, in the primed structure: The interaction between Loop 2 and subunit 1 of actin (Fig 1f); the salt bridge between R219 and E442 that keeps the backdoor closed (Fig. 3j); etc.
2. For Fig 3j,k there is no fitting within the cryoEM map - how do we judge the accuracy of the structure as depicted?
3. The authors should provide more discussion of these results in relation to previous myosin 5 structural work (Sweeney and Houdusse labs).
4. I am concerned about the lack of attention to some details. Ext Data Fig 3a-d: The data for the inset traces do not match the data in the graphs. For example in Fig3b, either the kobs is incorrect (should be ~17 s⁻¹ for 20uM deac-amnio ATP) or the stopped flow schematic should say ~10uM not 20uM. The stopped flow schematic also has a typo "dec-amino ADP" instead of "deac-amino ADP". Insets for a-c appear to have either the wrong substrate concentration or the wrong kobs. For 3d, the 89uM actin point is not shown on the graph.
5. They are possibly overstating their ability to claim that Pi is released before the stroke – their time resolution does not seem good enough.

Minor comments:

1. Fig 1a - the light blue densities in the actin monomers are not identified in the figure legend (presumably nucleotide)
2. Fig 5 legend - end-on and top views swapped/mislabeled

Referee #3

(Remarks to the Author)

Myosin motors couple ATPase activity with actin filament binding and release to generate forces which support a wide variety of critical cellular processes. Numerous structures of multiple myosin classes have been solved, representing the majority of F-actin bound and unbound states relevant to the motor's mechanochemical cycle. However, one key state has been missing: the "actomyosin primed state", when the motor initially contacts F-actin while still bound to ADP and inorganic phosphate, prior to the powerstroke. Despite its obvious importance, obtaining this structure has been exceptionally technically challenging. Under the equilibrium / steady state conditions accessible by traditional structural biology methods, this state is extremely rare since rapidly transitions to the post-powerstroke state. Therefore, while the actomyosin primed state has been postulated for decades, it has never been experimentally visualized.

In this study, the authors employ the emerging method of time-resolved cryo-EM to capture this elusive, transient state. In addition, they also report structures of unbound myosin in the primed state (present in the background of their images), as well as the post-powerstroke state captured after a longer time delay between mixing and freezing. Comparing these structures revealed transitions from the primed state to the post-powerstroke state, allowing direct visualization of myosin's actin-activated lever arm swing for the first time, thereby reconciling decades of speculation and indirect biophysical evidence. This study therefore fills a major gap in our understanding of myosin's mechano-chemical cycle, and it represents a long-awaited breakthrough in the myosin field suitable for Nature as its venue.

Despite the work's fundamental importance, we have a few substantive concerns and suggestions which we believe should be addressed in a revised manuscript prior to acceptance.

Major points:

1. The observation of the actomyosin in the primed state is truly exciting. However, a number of the detailed interactions claimed are not unequivocally supported by the density map at the current resolution. For example, in Fig. 1H the density of

the N-terminus of actin doesn't support the accurate assignment of the side chains of actin residues 1-4. Similar claims are made in the interpretation of the postPS density. In Fig. 3h-i, the detailed interactions of actin's D1 and E2 with myosin and the corresponding structural changes are more speculative than definitive. The authors should tone down the claims a bit.

2. The authors should mention the major differences between different actin isoforms at the protein's N-terminus. The skeletal actin used in this study is not the physiological track for myosin V, which should be a mixture of cytoplasmic β and γ -actin. It will be helpful to discuss these differences due to the actin N-terminus's essential role in myosin activation.

3. Most of the conformational changes the authors discuss are substantial, and well-resolved at 4-5 Angstroms resolution, where the protein backbone and a few large side chains are clearly visible. They are generally to be commended for their balanced presentation given the resolution limitations. However, this breaks down a bit in Figure 2 (comparing primed unbound myosin with primed actomyosin), where the changes are quite subtle. It would be helpful if the authors could show a morph of the segmented myosin density between these two maps in a supplemental movie, in support of the "cocking" rearrangement displayed in Fig. 2a-c. The subtle rearrangement of the relay helix (Fig. 2d-e) does not appear to be well-supported upon inspection of the density maps. However, rearrangements in the vicinity of the nucleotide cleft (Fig. 2f-h) are consistent with the maps.

4. From a kinetic point of view, is there a specific reason to choose the 120ms time point to capture the PostPS state? A related question is why there is no population of the rigor (nucleotide free) state, since the mutant myosin used in this study displays a 2 fold increase in ADP release rate compared to WT? Is this due to very rapid ATP rebinding?

5. A recent study (Pospich et al., 2021) showed that the lever arm undergoes a 9° rotation from the strong ADP bound state to the rigor state. In the present study, the lever arm position of the postPS state (which should be equivalent to Pospich et al.'s ADP-bound state) resembles Pospich et al.'s reported structure of the rigor (nucleotide free) state, likely due to the introduced mutations as suggested by the authors. In this regard, a structure of the mutant actomyosin in the rigor state would help clarify the ambiguity. If the ADP and rigor state of the mutant actomyosin are indeed highly similar, the phrasing of the proposed model would need to be adjusted. In this case, the structural comparison presented here actually represents the transition from the primed state to the rigor state, while the role of the ADP state remains to be determined.

6. The limited resolution of the actin-bound reconstructions, particularly the post-power stroke structure, is somewhat surprising given the number of particles (~100,000) included in the final reconstructions. One potential limitation is ice thickness, which the authors allude to obliquely. It would be useful to discuss this explicitly in the text.

Another potential limitation is continuous conformational variability, particularly of the primed actomyosin state. Analyzing the data with modern variability approaches (e.g. cryoSPARC 3DVA or cryoDRGN) would likely be informative. It might also enable the authors to hone in on a subpopulation that could be refined to moderately higher resolution.

Minor points:

7. The authors termed the two myosin contacting actin protomers as -actin and +actin. We suggest naming the actin protomers numerically (such as i, i+1 and i-1), adopting the convention of most other actomyosin publications, which would make the paper easier to understand for readers familiar with this literature

Version 1:

Reviewer comments:

Referee #2

(Remarks to the Author)

The authors added more data to make the story more solid. The responses to all my comments are reasonable, assuming that the use of MD simulation to model the side chain interactions and make their claims are common practice, as they say. I now recommend the paper for publication

Referee #3

(Remarks to the Author)

In their revised manuscript, the authors were responsive to the comments but some of the underlying scientific issues remain. A common critique by all three reviewers was that the resolution of the maps is too low to make definitive claims about side chain positioning, which in our view was not necessary for the most important take-home messages of the paper.

While the authors have now emphasized they are using MDFF for fitting in the text, none of the original claims were modified. This also extends to the putative subtle repositioning of the relay helix noted in our original review. Furthermore, new text has also been added claiming, "Intermediate states between the primed and postPS states were not detected despite extensive 3D classification and masking (Extended Data Fig. 4), indicating that the lever swing mechanism is a rapid, continuum structural change, analogous to firing a Roman catapult, rather than comprising a series of discrete steps." (lines 138-142). This is, in our view, logically problematic. The inability to identify intermediates using a particular classification approach does not mean they do not exist: there are many reasons classification can fail. Unfortunately, it is unclear whether our suggestion to try different variability approaches was implemented. The authors say they tried to use multibody refinement, which is not likely to be the most appropriate in this case, as the moving entity is quite small (the lever arm and converter) relative to the overall size of the motor domain. Thus, a more accurate statement is that no intermediate states could be detected using current classification approaches, consistent with a continuum, although they may be resolved in the future. Actually visualizing the continuum with e.g. 3DVA or cryoDRGN would be more convincing. On a more positive note, the inclusion of the rigor actomyosin structure, as was requested, does strengthen the argument that the mutant motor they are using has a standard mechanochemical cycle. We believe this is an important addition. Furthermore, the authors have done a nice job emphasizing the significance of the primed structure. We stand by our initial assessment that this is a very important structure that fills a key gap in understanding the myosin mechanism.

Referee #4

(Remarks to the Author)

The manuscript by Klebl et al. focus on the interaction between myosin and F-actin using time-resolved cryoEM studies to locate key points in the mechanism of force generation. The manuscript appears to be interesting, reporting innovative work and impactful results. I am neither an expert on the biological problem nor on the cryoEM methodologies used here. I was asked to comment on the use of MD in the interpretation and "refinement" of the models resulting from experimental data, where side-chain conformations and interactions are inferred and used in mechanistic explanations. Therefore, my comments will be focus in this.

The authors do not provide explanations in the manuscript about the MD refinement methodology used to infer side-chain conformations and interactions on the cryoEM derived models. Instead, they quote a paper by Scarff et al. 2020. I went through this paper and found out that the simulations used to generate side chain conformations and interactions consisted in AMBER14SB GBSA MD simulations with backbone restrained (how? No data is provided) proteins, lasting for 3ns to "allow formation of ionic interactions.". They also claim to use ISOLDE for certain parts, a method that performs similar type of calculations, although the details are missing. The "refinement" of sidechain conformations and interactions is important for the conclusions put forward by the authors, highlighting the importance of doing this as best as possible. My comments are the following:

- 1) The methodologies should be explicitly described in the present manuscript, for proper data reproduction.
- 2) Although GBSA simulations are likely to explore more conformational space than normal explicit solvent simulations with similar time spans, due to absence of solvent drag, 3 nanosecond simulations seem to me as too limited for obtaining reliable conclusions. I would prefer to see longer simulations, with independent replicates (3, at least) and, possibly in explicit solvent conditions. The statistics about the interactions (the claimed 50% of these) should be taken by analysing the replicates and not by analysing one sole simulation, which is known to be a significant source of errors and substandard in today's computational capabilities.
- 3) One would even improve the analysis further by allowing protein backbone flexibility, in areas where the experimental data does not warrant a proper description of the main fold. But this is debatable.
- 4) In the answer to reviewer #1, the authors claim that their MD refinement has been used before. While the validity of a method does not rely on its prior use, the authors are not consistent in the defence of their methodologies. They quote the work by Scarff et al. 2020 (the reference quoting the methodology), Bunduc et al. 2021 (where only ISOLDE was used), Dutta et al. 2023 (where, as far I could perceive, MD refinement in the context of cryoEM modelling is used; like in many experimental structural biology methodologies) and Pronker et al. 2023 (where free (with some parts restrained) explicit solvent MD at microsecond scales was employed). Therefore, the arguments towards the use of their methodology are misleading.

In conclusion, I think the work done by the authors has merit overall, but they should repeat the MD "refinement" following what I suggest above, specially the question of independent, longer time-spanning replicates, to infer about the statistical significance of the important interactions described. If possible, explicit solvent simulations should be used.

Version 2:

Reviewer comments:

Referee #3

(Remarks to the Author)

In their re-revised manuscript, the authors have addressed the remaining major concerns identified in the previous round of review. Notably, they have tempered or removed claims about sidechain positioning and subtle rearrangements, peripheral to the main message of the paper, that were weakly supported by the experimental data. They have also modulated their statement about the lack of intermediate states, which I believe now treats the implications of their classification analysis appropriately. The expanded molecular dynamics are also an excellent addition which strengthens their claims about the key new interactions identified in the primed state, the major take-home of this paper.

I believe this is now suitable for publication in Nature, which is an appropriate venue for the exciting and important structure of the actomyosin primed state.

Referee #4

(Remarks to the Author)

The authors of this manuscript revise it to clarify some points raised in the previous referee report and provide new simulations that enrich the work and solidify the conclusions put forward. I am convinced that the manuscript is now ready for acceptance.

Manuscript 2024-01-00287A-Z

Klebl et al.

Swinging lever mechanism of myosin directly demonstrated by time-resolved cryoEM

Authors' responses to the referees' comments

Dear Referees,

We thank you for your efforts in reviewing our manuscript.

We show here in full the remarks that were passed along to us by the Editor from yourselves, together with our responses (in blue). The line numbers refer to those in the manuscript with tracked changes for ease of reference.

Referee expertise:

Referee #1: time-resolved cryo-EM

Referee #2: actin/myosin, mechanistic

Referee #3: actin/myosin, cryo-EM

Referees' comments:

Referee #1 (Remarks to the Author):

The manuscript by Kelbl [Klebl] et al presents a time-resolved cryo-EM study of the interaction of F-actin with myosin-V and its swinging lever mechanism. This is a fundamental biological process with very wide relevance to basic and translational science and it has been studied in great detail over many decades, in particular also by molecular biology, biophysical and structural means. The authors correctly point out the lack of insights into short-lived, transient on-pathway structural intermediates.

We thank the referee for acknowledging the wide relevance to basic and translational science of understanding myosin structural intermediates to elucidate function. We focus here on one specific intermediate: the structure formed when primed myosin first binds to actin, before it has executed its working stroke. This had never been imaged before.

and thus attempt to address this by rapid mixing and snap-freezing of mixtures of a (slow hydrolysis) mutant form of primed myosin-V and F-actin at 10 and 120 ms time points. From a purely engineering perspective, this is no small achievement.

We thank this referee for acknowledging the technical advances that our work represents.

However, the obtained biological insights are very limited.

Here, we and others disagree. As Referee #3 points out: "This study therefore fills a major gap in our understanding of myosin's mechano-chemical cycle, and it represents a long-awaited breakthrough in the myosin field..." Moreover, the importance of these results were recently highlighted in *Nature*, as part of a News feature (<https://www.nature.com/articles/d41586-024-00817-y>) based on the draft manuscript that we deposited within bioRxiv (<https://www.biorxiv.org/content/10.1101/2024.01.05.574365v1.article-metrics>).

We are grateful to this referee for making us aware that we had not made plain the significance of our new data, as it could have easily escaped other readers of *Nature* who are not specialists in myosin motors. We have therefore revised the whole manuscript to signpost more clearly the advances and insights that our new data permit.

The authors were able to determine only two structures at resolutions just high enough to warrant tentative molecular interpretation. One of these represents a post power stroke actomyosin-V conformation, as is admitted by the authors, basically reproduces one of the dozens trapped structures of the pathway reported by the Raunser department (Pospich et al., eLife, 2021). In effect, this work reports a single novel structure,

Yes, we do indeed report a single novel structure, the complex formed when myosin-ADP-Pi initially binds to actin that has eluded all previous attempts to observe it, and it is key to understanding how all myosin motors work. We could have written the paper entirely about this one structure, because it fills a major gap in our understanding of myosin's mechano-chemical cycle. However, we took the opportunity to compare it with structures that we were able to obtain from the same sample, namely the primed detached myosin and the postPS actomyosin.ADP structures. We did this to enable direct comparisons as to how our novel structure fits within the mechano-chemical cycle. The postPS actomyosin structure is described as such as it is akin to the strongly-bound states reported in Pospich et al., which we state as follows:

Line 81 'Actomyosin structures in the ADP and nucleotide-free states, obtained by cryo electron microscopy (cryoEM), reveal the architecture of strongly-bound actomyosin complexes in which both the U50 and L50 subdomains interact with actin, the cleft is closed, and the lever adopts a post-powerstroke (postPS) position¹⁵.'

We have added further clarification of this from line 135, 'The postPS actomyosin structure was *satisfyingly* similar to previous structures of *myosin-5* strongly-bound states¹⁵, as expected given the mutant we used is fully functional in motility assays and only has a 2-fold increase in ADP release rate compared to the wild-type motor¹⁹'

... which falls far short of the claimed elucidation of the power stroke mechanism.

We disagree. By observing this missing intermediate and comparing it with the states formed prior to and after it in the mechanochemical cycle, we can elucidate how the powerstroke mechanism occurs.

As such, this manuscript would be of interest to researchers specialized in the structural biology of myosin and might be suitable for a specialized structural journal in a short report form. Even so, there are quite some major points that the authors would need to work on before it would be acceptable for publication. Thus, should a direct transfer to NSMB be of interest, careful attention should be paid to the below major points:

We hope that the revised version of this manuscript, alongside consideration of the remarks of the other referees, convince this referee that this work is a long-awaited breakthrough in the myosin field, deserving publication in *Nature*.

1. The claims in the manuscript are very inflated. At global resolutions of 4.4 and 4.9 Å and notably lower local resolution especially in functionally relevant loop regions, most of the modelled side chains and bonds are to be taken with the utmost caution.

We strongly refute this comment. The main claims in the manuscript relate to showing how myosin initially interacts with actin in its primed state, through the L50 helix-loop-helix region, with no U50 contacts, and how it then transitions to a post-powerstroke state, directly

demonstrating the swinging lever mechanism. These claims relate to gross conformational change of entire domains that are clearly resolved and do not rely on specific side chain interactions.

Nevertheless, at the resolutions achieved we can accurately place the backbone chain and for the great majority of the structure, the main chain conformation is so like pre-existing higher resolution structures that we can reasonably use the rotamers from those structures, and we have therefore done so. Where use of existing structures for side chain placement was not sensible, especially at new contact interfaces identified in this study, we have modelled side chains based on map density and the use of MD simulations. This is a common and established practice in the cryoEM field and was used effectively in many recent studies published in *Nature* to produce plausible pseudo-atomic models, e.g., Scarff *et al.* 2020 (6 Å global resolution), Bunduc *et al.* 2021 (4 Å global resolution), Dutta *et al.* 2023 (6 Å global resolution), Pronker *et al.* 2023 (3-4 Å global resolution). We apologise that this may not have been made clear within our Methods section and we have added description of this into the main text.

We have inserted at line 129, 'To enable detailed interpretation of the cryoEM density maps, we followed current practice²²⁻²⁵ to create pseudo-atomic models by flexibly fitting available high-resolution structures into the maps, performing homology modelling and refinement and using molecular dynamics simulations to explore side chain interactions at contact interfaces (see Methods, and Extended Data Table 1 & 2).'

More importantly, the lack of any intermediate structures from primed actomyosin state to the post power stroke conformation, which is not novel but rather reproduces a conformation reported in 2021, should effectively force the authors to discuss their two structures in the context of the wealth of other structural work.

The lack of intermediate structures is not a deficiency of the study but is itself a new observation. If discrete intermediate structures were sufficiently populated along the pathway, then our dataset would contain them. That suggests that the huge conformational changes of the myosin working stroke happen as a continuum structural change, like in a catapult. In response to this referee's remark, we have placed greater emphasis on the absence of discrete structural intermediates in the revised manuscript, and have introduced the analogy of the catapult to bring home to the general reader the type of mechanism that happens in myosin's working stroke.

We have inserted at line 138, 'Intermediate states between the primed and postPS states were not detected despite extensive 3D classification and masking (Extended Data Fig. 4), indicating that the lever swing mechanism is a rapid, continuum structural change, analogous to firing a Roman catapult, rather than comprising a series of discrete steps.'

We are aware that the postPS structure is not novel and we do say so, mentioning how it compares to previous structures, Line 135 'The postPS actomyosin structure was satisfyingly similar to previous structures of myosin-5 strongly-bound states¹⁴, as expected given the mutant we used is fully functional in motility assays and only has a 2-fold increase in ADP release rate compared to the wild-type motor¹⁸.' and line 372 'Our postPS actomyosin-5 structure shows a closed actin-binding cleft as well as a postPS lever position (Fig. 3d) and has high similarity to previous structures of strongly-bound actomyosin complexes (ADP-bound or rigor states)¹⁴.'

We compared the primed actomyosin structure with the postPS structure derived from the same sample to aid in our interpretation of how our new structure fits into the mechanochemical cycle.

Surprisingly, the authors instead present a very detailed molecular descriptions of molecular interfaces (including side chains in loops and salt bridges in areas of very low local resolution)

and conformational rearrangements by 2-3 Å (i.e. smaller than the global resolution of the reported maps). Such notes, at best, belong in an extended supplementary text's discussion and figures and should also include equally detailed comparison to published work.

In an EM density map, small shifts in position of electron density are a fundamentally different parameter from the resolution of the map itself and shifts can be detected that are much smaller than the resolution. To illustrate this point, we have amended Supplementary Video 2 within the revised manuscript to show a morph between the electron density maps of primed myosin-5 free in solution and bound to actin and signposted to this in both the text and appropriate figure legend. This clearly shows that domain shifts of 2-3 Å are visible within the EM density at 4.4-4.9 Å global resolution. As forementioned, we have fitted pre-existing high-resolution structures within the maps and used MD simulations to best place side chains where appropriate, which is a highly established practice in the field. We have now explained this in more detail within the text.

Inserted at Line 129, 'To enable detailed interpretation of the cryoEM density maps, we followed current practice²¹⁻²⁴ to create pseudo-atomic models by flexibly fitting available high-resolution structures into the maps, performing homology modelling and refinement and using molecular dynamics simulations to explore side chain interactions at contact interfaces'.

2. The authors worked with a double mutant of myosin-V which slows down hydrolysis. Even though it is understandable that they had to resort to such mutants, because the studied reaction would be faster than their technically possible time resolution, it is not clear from the description in the paper whether these mutants alter the molecular pathway. The authors would need substantial backing from orthogonal biophysical assays to ascertain that any trapped intermediates (i.e. presently the single one) is consistent with the wild type pathway.

The mutant myosin we used is extensively characterised kinetically in our manuscript (Extended Data Fig. 3) and the effects of the loop2 deletion and switch-1 mutation on myosin activity have been characterised previously (Yengo & Sweeney, *Biochemistry* 2004, and Forgacs et al., *JBC* 2009, respectively, both cited in our manuscript). While the mutant used slows actin-activated Pi release approximately 10-fold and increases the affinity of the myosin-5-ADP-Pi primed state for F-actin ~10 fold, there is no evidence that it changes the molecular pathway, and it is fully functional in motility assays. While there are subtle structural differences in the mutant (that we explain) this does not change the main interactions and conclusions of the paper – binding is initially through the L50, followed by cleft closure and lever swing.

To resolve this comment fully, we have added the cryoEM data for rigor actomyosin of this mutant to the manuscript. This cryoEM density map is well accommodated by the motor domain of a previously solved rigor actomyosin structure (PDB ID 7PLV), such that it is consistent with the wild-type myosin-5 rigor actomyosin structure, with a slightly different lever/converter position to the ADP-bound actomyosin state.

We have inserted an explanation for this into the text at line 384 'To ensure this mutant motor still transitioned through the canonical mechanochemical cycle, with the expected second smaller lever displacement seen upon ADP release¹⁴, we also obtained a cryoEM density map of the mutant rigor actomyosin to a global resolution of 3.9 Å (Extended Data Fig. 10). This showed the expected lever and converter displacement on ADP release (Extended Data Fig. 11), indicating that this motor undergoes a classic mechanochemical cycle.'

Accordingly, we have added information to the methods on how this rigor actomyosin-5 mutant structure was obtained.

At line 599, we have inserted the following paragraph pertaining to grid preparation:

'Rigor actomyosin cryoEM grid preparation

3µl of pre mixed F-actin and myosin (1:1 ratio with a final concentration of 1 µM) was applied to a Quantifoil R2/2 300 mesh grid that had been glow discharged in a Cressington 208 Carbon coater for 30 s at 0.1 mbar air pressure and 10 mA prior to use. Grids were prepared by use of a Vitrobot Mark IV (ThermoFisher), with blot time of 6 seconds and blot force of 6 at 4 °C and 95 % humidity with vitrification in liquid ethane.'

At line 639, details of data processing:

'Processing parameters for rigor actomyosin-5 are listed in Extended Data Table 4 with an overview of the processing pipeline shown in Extended Data Fig. 10. Particles were manually picked from a subset of micrographs and after one round of 2D classification, good particles were used to train a crYOLO model⁴⁸. With the trained model, particles were picked from the entire rigor dataset, leading to a final selection of 85986 particles after one round of 2D classification. The particles then underwent two rounds of 3D helical refinement and Bayesian polishing. Following particle polishing, signal outside the masked actomyosin complex (myosin and the central 3 actin subunits) was subtracted. The subtracted particles then underwent two rounds of focused classification on the myosin motor domain leading to a final selection of 21890 selected particles. The final 3D refinement was then done using non-uniform refinement in cryoSPARC⁴⁹.

At line 664, details of modelling: 'To enable interpretation of our mutant rigor actomyosin-5 cryoEM density, we rigidly fitted existing rigor actomyosin-5 PDB structures generated by Pospich et al¹⁴, into the map. The myosin chain from PDB ID 7PLV was well accommodated into the map, especially within the converter region (Extended Data Fig. 11f). As such we used this as a model for our rigor myosin-5 structure to enable us to compare it to our PostPS actomyosin-5 structure (Extended Data Fig. 11f).'

3. The paragraph on "structural mechanism of myosin force generation and ATPase activation on F-actin" is not appropriate in the results part as it is based on a single novel structure. It is a well-written and important piece of text but much rather suitable for a review article.

We are glad that the referee finds our text important, but we disagree fundamentally with the opinion that it belongs in a separate publication. Having fitted the last piece into the jigsaw puzzle it is entirely proper that we describe the picture that we can now see, and how the new piece interacts with the surrounding pieces to create the complete picture.

In summary, this manuscript attempts a technically difficult time-resolved structural study of an important pathway but falls short of delivering any novel biological insights

We strongly disagree – we show how myosin initially interacts with actin prior to the powerstroke and its turnover to the postPS state, directly demonstrating the swinging lever mechanism (recognised by the other two referees and the opinion piece in Nature). We have revised the wording of the paper throughout so that the novel biological insights we reveal are more clearly signposted.

... and presents very modest experimental results relative to the claims made. In order for the manuscript to be accepted in a peer-reviewed journal, it would be important to improve the resolution to a point that allows the authors to interpret the structure(s) at near-atomic resolution, in particular in the regions that appear flexible at the 10 ms time point

Here, we are not dealing with small rotamer structural changes producing ion transport within a membrane pore protein, we are dealing with a motor protein in which whole subdomains move by nanometres. As such there are many new insights that we enumerate from the structure we now have that have only been speculation in the past, and others that have never before been considered, as indeed the other two referees emphasise. Yes, in time we aim to push the structures to higher resolution, as additional insights will undoubtedly emerge, but that is for the future. What we have done here is to provide the first structure of the actomyosin primed state, 'a long-awaited breakthrough in the myosin field' (reviewer #3).

... as well characterize the mutant they used. Moreover, for the work to be of interest to a broad readership of a specialized and highly respected journal like NSMB or Nat Comms, it would be important to attempt to solve further previously unknown states, comment what might be in the way of observing most of the intermediates, and thoroughly contextualise the results with respect to the wealth of previously reported structural intermediates.

Our empirical evidence is that the lever swing does not have any (other) intermediate states. There are no biochemical or structural studies by us or other groups that show additional states are present between primed and strong-ADP actomyosin states. A Pi-release state has been proposed but not demonstrated on actin. We do refer to studies that have focussed on other stages of the cycle (e.g. rigor, off-actin myosin structures), but focus here on our results and how they add to our understanding of myosin's mode of action. In addition, we now refer the reader to the very thorough review by Robert-Paganin et al. on the current understanding of force generation by myosin motors, from a structural perspective at line 76, 'The current understanding of force generation by myosin motors, based on the available structural states to date, is reviewed in detail in Robert-Paganin et al.³'

Referee #2 (Remarks to the Author):

The manuscript "Swinging lever mechanism of myosin demonstrated by time-resolved cryoEM" by Klebl et al uses time resolved cryoEM and a myosin 5 mutant with slowed Pi release and increased actin affinity to capture a 4.4Å resolution structure of the primed actomyosin complex, which has previously not been visualized. The approach is novel, the experiments well conceived, and the manuscript well written. I believe it is of sufficiently broad interest to warrant publication in Nature with the following issues addressed.

We are grateful for these very positive comments from Prof. Spudich.

1. Claims involving side chain interactions are not warranted – the resolution of the map does not allow one to see individual side chains. This needs to be recognized when reporting, for example, in the primed structure: The interaction between Loop 2 and subunit 1 of actin (Fig 1f); the salt bridge between R219 and E442 that keeps the backdoor closed (Fig. 3j); etc.

As we mention in the manuscript, we used MD simulations to explore how the side chain rotamers would adjust in regions of contact between actin and myosin. While we do not have EM density to accurately position all the side chains, they are there in the specimen and so not to state potential interactions seems a disservice to the reader. However, to address Prof Spudich's concerns, we have modified the text to make it clearer that our discussion of side chain interactions is based on previous studies and MD simulations, rather than direct observation of side chain rotamers in our new structure.

e.g. Line 176, 'MD fitting of side chains suggests that this...' and line 190, 'Modelling in ISOLDE showed that.'

Specifically for the salt bridge between R219 and E442 within the myosin motor domain, there is a big movement of the polypeptide backbone between the two states: the main chain atoms

of these residues are close enough in the primed actomyosin to allow a strong interaction between the side chains to be found by MD, as has of course been amply demonstrated in other studies of primed myosin not bound to actin. In the postPS structure, the backbones are further apart and the side chains are not likely to interact, as others have previously shown. Therefore, we feel justified in concluding that this salt bridge is present at the start of the working stroke and absent at the end.

We have amended the text accordingly to read at line 294, 'Switch-1 and switch-2 loops are in such close proximity in the density map that in our model the salt bridge between R219 in switch-1 and E442 in switch-2, termed the backdoor, is still present and so the exit door is closed.', and at line 338 'salt bridge... is intact within our primed actomyosin pseudo-atomic model and broken in our postPS model, consistent with previous structural observations^{3,12,15}.'

2. For Fig 3j,k there is no fitting within the cryoEM map - how do we judge the accuracy of the structure as depicted?

We have added the density map to Figure panels 3j,k, however, given the complexity of the density in this region it is hard to ascertain how the model fits within it. This is really only possible to appreciate when viewing the density map and fitted model in 3D, as such we now suggest that the reader does just that. We have inserted at line 369 '(see Supplementary Video 3 and EM density maps).'

3. The authors should provide more discussion of these results in relation to previous myosin 5 structural work (Sweeney and Houdusse labs).

There is a very large body of previous structural work on myosin examining force generation, performed by a variety of groups, which cannot all be discussed within this manuscript. We have thus suggested in the text that the interested reader should refer to Robert-Paganin et al. (Force Generation by Myosin Motors: A Structural Perspective. Chemical Reviews, 2020) a thorough review on the topic, by inserting at line 76, 'The current understanding of force generation by myosin motors, based on the available structural states to date, is reviewed in detail in Robert-Paganin et al.³'

4. I am concerned about the lack of attention to some details. Ext Data Fig 3a-d: The data for the inset traces do not match the data in the graphs. For example in Fig3b, either the kobs is incorrect (should be ~17 s⁻¹ for 20uM deac-amnio ATP) or the stopped flow schematic should say ~10uM not 20uM. The stopped flow schematic also has a typo "dec-amino ADP" instead of "deac-amino ADP". Insets for a-c appear to have either the wrong sustrate concentration or the wrong kobs. For 3d, the 89uM actin point is not shown on the graph.

The misunderstanding here is that the values stated in the insets are the concentrations of reactants in the syringes not in the final reaction mixture. These get diluted by the contents of the other syringes when mixed to produce the final reaction mixture. Once that is appreciated, there are no errors in the Figure, apart from the typo, for which we apologise. Thus, for Fig 3b, the final concentration of deac-aminoADP is 10 μ M, and the measured rate of 11.7/sec is correctly plotted at the 10 μ M concentration. Again, for 3d, the 89 μ M actin is diluted twofold to 44.5 μ M, and the measured rate is correctly plotted on the actin axis.

This depiction of syringe concentrations is common practice in describing stopped flow kinetics experiments but will not be familiar to more general readers. We have therefore added a sentence to the legend to make this clear, which states:

'Representative traces shown in insets with a stopped-flow mixing schematic. Note protein concentrations stated are those within the syringes, prior to mixing, rather than in the final reaction mixture.'

We have also added text to explain that in Fig 3d the volume of actin used in the second push is equal to the volume of the myosin-ATP mixture, such that there is a twofold, not threefold,

reduction of actin concentration. '2 μM myosin was mixed with 1.6 μM ATP, held in a delay line for 2 s, and then mixed with equal volume of actin to accelerate P_i release.'

5. They are possibly overstating their ability to claim that P_i is released before the stroke – their time resolution does not seem good enough.

We propose a mechanism by which myosin produces movement. P_i release from the active site before the powerstroke is part of that proposal. However, since cleft closure must occur to strengthen motor binding to actin and thereby allow a productive powerstroke against a resisting force, and since during cleft closure the backdoor is opened enabling P_i release, we see no alternative to what we describe. We selected time points based on our kinetic analysis of myosin-5 in Extended Data Fig. 3. Incubation times were chosen to produce samples in which there was either minimal time for P_i dissociation (10ms) or an extended time (120 ms) to increase the fraction of molecules from which P_i had dissociated.

Minor comments:

1. Fig 1a - the light blue densities in the actin monomers are not identified in the figure legend (presumably nucleotide)

Prof. Spudich is correct – this is the nucleotide and we have now augmented the legend to state this. Thank you.

2. Fig 5 legend - end-on and top views swapped/mislabeled

We thank Prof. Spudich for his careful reading of the manuscript, and we apologise for these mistakes that were generated when we rearranged the panels to make a more compact, single column layout. They are now corrected.

Reviewer: Jim Spudich

Referee #3 (Remarks to the Author):

Myosin motors couple ATPase activity with actin filament binding and release to generate forces which support a wide variety of critical cellular processes. Numerous structures of multiple myosin classes have been solved, representing the majority of F-actin bound and unbound states relevant to the motor's mechanochemical cycle. However, one key state has been missing: the "actomyosin primed state", when the motor initially contacts F-actin while still bound to ADP and inorganic phosphate, prior to the powerstroke. Despite its obvious importance, obtaining this structure has been exceptionally technically challenging. Under the equilibrium / steady state conditions accessible by traditional structural biology methods, this state is extremely rare since rapidly transitions to the post-powerstroke state. Therefore, while the actomyosin primed state has been postulated for decades, it has never been experimentally visualized.

In this study, the authors employ the emerging method of time-resolved cryo-EM to capture this elusive, transient state. In addition, they also report structures of unbound myosin in the primed state (present in the background of their images), as well as the post-powerstroke state captured after a longer time delay between mixing and freezing.

Comparing these structures revealed transitions from the primed state to the post-powerstroke state, allowing direct visualization of myosin's actin-activated lever arm swing for the first time, thereby reconciling decades of speculation and indirect biophysical evidence. This study therefore fills a major gap in our understanding of myosin's mechanochemical cycle, and it represents a long-awaited breakthrough in the myosin field suitable for Nature as its venue.

We thank this referee for their accurate and supportive assessment of our paper.

Despite the work's fundamental importance, we have a few substantive concerns and suggestions which we believe should be addressed in a revised manuscript prior to acceptance.

Major points:

1. The observation of the actomyosin in the primed state is truly exciting. However, a number of the detailed interactions claimed are not unequivocally supported by the density map at the current resolution. For example, in Fig. 1H the density of the N-terminus of actin doesn't support the accurate assignment of the side chains of actin residues 1-4. Similar claims are made in the interpretation of the postPS density. In Fig. 3h-i, the detailed interactions of actin's D1 and E2 with myosin and the corresponding structural changes are more speculative than definitive. The authors should tone down the claims a bit.

The atomic model side chains do fit into lobes of density (this is especially clear in Fig. 3h, i), and the assignment is supported by the results of MD simulation, so we would be surprised if our conclusions are found to need revision in the future. Nevertheless, we take on board the referee's advice and have gone through the manuscript making the wording of our conclusions more tentative where appropriate.

2. The authors should mention the major differences between different actin isoforms at the protein's N-terminus. The skeletal actin used in this study is not the physiological track for myosin V, which should be a mixture of cytoplasmic β and γ -actin. It will be helpful to discuss these differences due to the actin N-terminus's essential role in myosin activation.

This is an excellent point and we have taken the opportunity of the revision to briefly highlight this gap in knowledge of the role of actin isoforms in myosin function.

At line 187, we have inserted 'DEDE for skeletal alpha actin used here, and -DDD, -EEE for beta and gamma actins respectively' and at line 416 'It is important to note that myosin-5's natural actin substrate is cytoplasmic actin (beta and gamma), rather than the skeletal alpha actin used here, and that these actins have N-termini with slightly different sequence but conserved negative charge such that the interactions reported here are likely maintained.'

We would also like to point out that to our knowledge, all previous in-vitro work on myosin-5's interactions with actin has been done using the striated muscle (α) isoform of actin and the properties thus revealed are consistent with the behaviour of myosin-5 in the cell.

3. Most of the conformational changes the authors discuss are substantial, and well-resolved at 4-5 Angstroms resolution, where the protein backbone and a few large side chains are clearly visible. They are generally to be commended for their balanced presentation given the resolution limitations.

Thank you!

However, this breaks down a bit in Figure 2 (comparing primed unbound myosin with primed actomyosin), where the changes are quite subtle. It would be helpful if the authors could show a morph of the segmented myosin density between these two maps in a supplemental movie, in support of the "cocking" rearrangement displayed in Fig. 2a-c.

We entirely agree that these movements are subtle and hard to show in static 2D Figures and so have inserted additional video into the revised manuscript (a revised Supplementary Video 2). This now shows the morph between the primed myosin and primed actomyosin EM density

maps, aligned on the HLH motif, before then progressing to show the morphing between the fitted atomic models.

The subtle rearrangement of the relay helix (Fig. 2d-e) does not appear to be well-supported upon inspection of the density maps. However, rearrangements in the vicinity of the nucleotide cleft (Fig. 2f-h) are consistent with the maps.

The subtle rearrangement of the relay helix we observe comes from directly overlaying the models on residues 449-457 at the N-terminal end of the relay helix. This difference is not obvious from direct comparison of the maps, as the reviewer comments. As the fulcrum of the cocking back of the U50 relative to the L50 is close to the relay helix, the super-kinking of the relay helix is small, but it is consistent with the global changes in domain arrangement that we observe. We have now discussed this point in more detail within our manuscript.

We have inserted at Line 253 'This super-kinking of the relay helix is rather subtle, and only observed when the two models are directly overlayed on residues 449-457. This is because the fulcrum of the cocking back of the U50 relative to the L50 is close to the N-terminal end of the relay helix and as such this subtle change is consistent with the global changes in domain arrangement that we observe.'

We thank the reviewer for stating that rearrangements in the vicinity of the nucleotide cleft are consistent with the maps.

4. From a kinetic point of view, is there a specific reason to choose the 120ms time point to capture the PostPS state? A related question is why there is no population of the rigor (nucleotide free) state, since the mutant myosin used in this study displays a 2 fold increase in ADP release rate compared to WT? Is this due to very rapid ATP rebinding?

As shown in Extended Data Fig 5, there are both primed and postPS particles in both the 10 ms and 120 ms timepoints, but with the key difference that their relative abundance switches over with time. So particles obtained in both the 10 ms and 120 ms datasets contributed to both actomyosin maps. We have amended the layout of Extended Data Table 1, and the legend, so this is clear, inserting 'Data from all six datasets was combined and processed to give rise to the primed and postPS actomyosin cryoEM density maps.'

As stated in the main text, we chose 120 ms simply to demonstrate the switch over (predicted by the 13.4 s^{-1} Pi-release rate, shown in Extended Data Fig. 3), reasoning that 120 ms was long enough for most working strokes to occur, but not so long that the mixture would be approaching steady state. The referee is correct that with $250 \mu\text{M}$ ATP in the mixture, both ATP rebinding and dissociation of myosin-ATP from actin are fast. We have now explained this in our Results to resolve the same query arising in the minds of other readers of the paper. We have inserted at line 142, 'As expected, rigor actomyosin was not identified in the time-resolved data as the high concentration of ATP in the mixture meant ATP rebinding and dissociation of myosin-ATP from actin would occur rapidly.'

5. A recent study (Pospich et al., 2021) showed that the lever arm undergoes a 9° rotation from the strong ADP-bound state to the rigor state. In the present study, the lever arm position of the postPS state (which should be equivalent to Pospich et al.'s ADP-bound state) resembles Pospich et al.'s reported structure of the rigor (nucleotide free) state, likely due to the introduced mutations as suggested by the authors. In this regard, a structure of the mutant actomyosin in the rigor state would help clarify the ambiguity. If the ADP and rigor state of the mutant actomyosin are indeed highly similar, the phrasing of the proposed model would need to be adjusted. In this case, the structural comparison presented here actually represents the transition from the primed state to the rigor state, while the role of the ADP state remains to be determined.

Our two structures have ADP in the nucleotide pocket, so the transition we describe is the working stroke of this mutant, primed to strong-ADP. Naturally, it would be preferable to work with wild-type myosin-5, and armed with the progress we report here, we hope to pursue that objective in a future study. We have also collected data for this mutant under rigor conditions. We had omitted this rigor structure from the manuscript to not overcomplicate the message, as it is not needed to interpret the actomyosin primed to strong-ADP transition. But we appreciate now that the referee's question will arise in the minds of many readers, and so we have added the rigor cryoEM density accordingly. We can show that the rigor cryoEM density map is accommodated well by one of the Pospich et al. rigor myosin structures (PDB ID: 7PLV chain A) and that in comparison to the PostPS structure there is a small kick of the lever upon nucleotide release of around 6°, which is within the order of variation seen in the multitude of structures reported in Pospich et al. 2021.

The details of the rigor actomyosin data collection and data processing pipeline have been added to the methods at line 599 and line 639 respectively and illustrated in Extended Data Table 4 and Extended Data Fig. 10. Comparison of the rigor and PostPS actomyosin structures is now discussed in the text and illustrated in Extended Data Fig. 11.

We have inserted at line 384, 'To ensure this mutant motor still transitioned through the canonical mechanochemical cycle, with the expected second smaller lever displacement seen upon ADP release¹⁴, we also obtained a cryoEM density map of the mutant rigor actomyosin to a global resolution of 3.9 Å (Extended Data Fig. 10). This showed the expected lever and converter displacement on ADP release (Extended Data Fig. 11), indicating that this motor undergoes a classic mechanochemical cycle.'

6. The limited resolution of the actin-bound reconstructions, particularly the post-power stroke structure, is somewhat surprising given the number of particles (~100,000) included in the final reconstructions. One potential limitation is ice thickness, which the authors allude to obliquely. It would be useful to discuss this explicitly in the text.

We thank the reviewer for pointing this out and agree that it would be useful to have more detail on the limitations on the overall resolution and have therefore inserted a brief paragraph into the Methods section on the factors that acted to limit resolution in our study, inserted at line 592, 'A typical feature of grids made by spraying is thicker ice compared to the use of more standard approaches⁴², such as the Vitrobot. This is due to the requirement for the drops to "thin" after deposition on the grid, which can be limited by the short time between spraying and vitrification. This increase in ice thickness will be a significant factor in limiting resolution to ~4.0 Å, alongside conformational flexibility¹⁴.'

Another potential limitation is continuous conformational variability, particularly of the primed actomyosin state. Analyzing the data with modern variability approaches (e.g. cryoSPARC 3DVA or cryoDRGN) would likely be informative. It might also enable the authors to hone in on a subpopulation that could be refined to moderately higher resolution.

We appreciate that conformational variability will limit resolution and we did perform extensive 3D classification and multibody analysis of the data for just the reason the referee suggests, to find a subpopulation that we could push to higher resolution. The primary aim was to search for discrete conformational changes rather than continuous flexibility and therefore focussed on 3D classification over approaches such as 3DVA. We didn't find any such class. We have clarified this in the appropriate Methods paragraph, at line 620 'particles were classified into actomyosin states or bare actin, with a small fraction of particles left unassigned. Despite performing multiple rounds of 3D classification, we found that actomyosin particles were classified into either a primed actomyosin state, with a primed lever and open actin-binding cleft, or a postPS actomyosin state, with a post-powerstroke lever and a closed actin-binding cleft (Extended Data Fig. 4c, e). No other actomyosin states were identified.'

Minor points:

7. The authors termed the two myosin contacting actin protomers as -actin and +actin. We suggest naming the actin protomers numerically (such as i , $i+1$ and $i-1$), adopting the convention of most other actomyosin publications, which would make the paper easier to understand for readers familiar with this literature

We encountered a significant problem with the recent literature on this point, which is why we have used the nomenclature that we do. The problem is the following. It has long been standard nomenclature that the barbed end of actin is the plus end and the pointed end the minus end, so subunit $i+1$ has naturally been the one nearer to the + end of the filament than subunit i . It is a convenient fact that most myosins move towards the + end of actin, so myosin-5 commonly steps to the actin subunit $i+13$ along the filament from the trail motor attachment. Then a paper appeared (Huehn et al., PNAS 2020, 117, 1478-84) in which the authors put the pointed (-) end of actin at the top of their Fig 1A and labelled the subunits in *decreasing* numerical order progressing downwards towards the barbed (+) end. That choice yielded the unfortunate result that subunit $i+1$ was nearer the pointed end rather than the barbed end. Sadly, subsequent papers have adopted this contradictory numbering system, such that it is now ambiguous what is meant – is subunit $i+1$ nearer the pointed or the barbed end of the actin filament? Since this question no longer has a clear answer, this nomenclature is 'broken'. We therefore feel forced to move away from this confused situation and refer to the actin subunit that is closer to the + (barbed) end as +actin and the actin subunit nearer the - (pointed) end as actin. We define these terms in the main text, but for improved clarity we have now also defined them in Fig. 1 legend.

Manuscript 2024-01-00287A-Z

Klebl et al.

Swinging lever mechanism of myosin directly demonstrated by time-resolved cryoEM

Authors' responses to the referees' comments

Dear Referees,

We thank you for your efforts in (re-)reviewing our manuscript.

We show here in full the remarks that were passed along to us by the Editor from the three referees from the second round of review, together with our responses (in blue). The line numbers refer to those in the manuscript with tracked changes for ease of reference.

Referees' comments:

Referee #2 (Remarks to the Author):

The authors added more data to make the story more solid. The responses to all my comments are reasonable, assuming that the use of MD simulation to model the side chain interactions and make their claims are common practice, as they say. I now recommend the paper for publication

Jim Spudich

We thank Prof. Spudich for his positive response. As you will see, in response to referee #4, we have now performed more thorough analysis of side-chain interactions by MD simulations that adds further confidence to our models and interpretation.

Referee #3 (Remarks to the Author):

In their revised manuscript, the authors were responsive to the comments but some of the underlying scientific issues remain. A common critique by all three reviewers was that the resolution of the maps is too low to make definitive claims about side chain positioning, which in our view was not necessary for the most important take-home messages of the paper.

We agree that side chain positioning is not necessary for the important take home messages of the paper.

As such, for the HLH-actin and loop3-actin interactions that are not new within our structures but conserved in other strongly-bound states of actomyosin-5, we have removed the discussion of side chain interactions from the text (at lines 172 and 186) and the associated panels from Figure 1 that showed these (Figure 1; line 208).

The side chain interactions that we suggest that are new within the primed actomyosin structure, are those between the actin N-terminal residues and myosin loop2 and helixW. We have now performed more thorough MD simulation analysis of these side chain positions, as requested by referee #4, which have added confidence to the positioning of these side chains within the model and we have modified the text at lines 181 and 200 to describe this as well as adding thorough description of the MD simulations used within the methods and new supporting Extended Data Figures (6 & 14) and Supplementary Videos (5 & 6).

At line 181, the text now reads' "MD driven fitting and subsequent simulation of side chain interactions suggest that positively charged residues K629 and K632 interact with the negatively charged D24 and D25, respectively in -actin subdomain-1 (Fig. 1e, Extended Data Fig. 6 and Extended Data Table 3).

At line 200, "Further examination of these modelled interactions, by explicit solvent MD simulations indicated that they are largely maintained over time (Extended Data Fig. 6, Extended Data Table 3)."

While the authors have now emphasized they are using MDFF for fitting in the text, none of the original claims were modified. This also extends to the putative subtle repositioning of the relay helix noted in our original review.

As discussed above, we have removed discussion of side chain interactions occurring at the HLH-actin and loop3-actin interfaces and performed additional MD simulations to add confidence to the positioning of side chains at the N-terminal actin myosin loop2/helixW interface. These additional simulations are fully described in the Methods, and two new Extended Data figures and Supplementary Videos have been added to the manuscript to illustrate the method and key results.

The subtle modification of the relay helix is what we observe upon comparison of the models, but we do not draw conclusions from it. As this is a subtle change, and we do not want to distract the reader from the main messages, we have removed this comparison from Fig. 2 and excised the associated description from the text (line 264).

Furthermore, new text has also been added claiming, "Intermediate states between the primed and postPS states were not detected despite extensive 3D classification and masking (Extended Data Fig. 4), indicating that the lever swing mechanism is a rapid, continuum structural change, analogous to firing a Roman catapult, rather than comprising a series of discrete steps." (lines 138-142). This is, in our view, logically problematic. The inability to identify intermediates using a particular classification approach does not mean they do not exist: there are many reasons classification can fail. Unfortunately, it is unclear whether our suggestion to try different variability approaches was implemented. The authors say they tried to use multibody refinement, which is not likely to be the most appropriate in this case, as the moving entity is quite small (the lever arm and converter) relative to the overall size of the motor domain. Thus, a more accurate statement is that no intermediate states could be detected using current classification approaches, consistent with a continuum, although they may be resolved in the future. Actually visualizing the continuum with e.g. 3DVA or cryoDRGN would be more convincing.

To try to detect intermediates, we carried out 3D variability analysis (3DVA) but this did not yield anything meaningful, likely because of particle number limitations. We consider that if any intermediate structures were present of similar abundance to the primed and postPS structures in our data, our methods would have detected it. We agree with the referee that intermediates, present in low abundance, could nevertheless exist, to be revealed in future studies, and are grateful for the referee's suggestion of an acceptable form of text to convey that uncertainty. We have therefore revised our text (line 142) to read: "Intermediate states between the primed and postPS states were not detected despite extensive 3D classification and masking" (Extended Data Fig. 4). This suggests that the lever swing mechanism may be a continuum structural change, analogous to firing a Roman catapult, but intermediate states present in low abundance could have escaped detection."

On a more positive note, the inclusion of the rigor actomyosin structure, as was requested, does strengthen the argument that the mutant motor they are using has a standard

mechanochemical cycle. We believe this is an important addition. Furthermore, the authors have done a nice job emphasizing the significance of the primed structure. We stand by our initial assessment that this is a very important structure that fills a key gap in understanding the myosin mechanism.

We thank the referee for acknowledging the addition of the rigor actomyosin structure, and that we have improved the communication of the key advances that our new structure represents.

Finally, we are gratified by the opinion of the referee that our primed actomyosin structure represents a key advance in understanding myosin's mechanism of action.

Referee #4 (Remarks to the Author):

The manuscript by Klebl et al. focus on the interaction between myosin and F-actin using time-resolved cryoEM studies to locate key points in the mechanism of force generation. The manuscript appears to be interesting, reporting innovative work and impactful results. I am neither an expert on the biological problem nor on the cryoEM methodologies used here. I was asked to comment on the use of MD in the interpretation and “refinement” of the models resulting from experimental data, where side-chain conformations and interactions are inferred and used in mechanistic explanations. Therefore, my comments will be focus in this.

The authors do not provide explanations in the manuscript about the MD refinement methodology used to infer side-chain conformations and interactions on the cryoEM derived models. Instead, they quote a paper by Scarff et al. 2020. I went through this paper and found out that the simulations used to generate side chain conformations and interactions consisted in AMBER14SB GBSA MD simulations with backbone restrained (how? No data is provided) proteins, lasting for 3ns to “allow formation of ionic interactions.”. They also claim to use ISOLDE for certain parts, a method that performs similar type of calculations, although the details are missing. The “refinement” of sidechain conformations and interactions is important for the conclusions put forward by the authors, highlighting the importance of doing this as best as possible.

Referee #3 states “...side chain positioning is not necessary for the most important take-home messages of the paper.” We agree with referee #3 on this point. Nevertheless, for the new side chain interactions that we suggest to occur between primed myosin and the N-terminal residues of actin in our new structure, and which could not be inferred from existing higher resolution structures, we accept that we needed additional MD simulation to strengthen our claims.

My comments are the following:

- 1) The methodologies should be explicitly described in the present manuscript, for proper data reproduction.

We thank the referee for identifying this omission and we have now explicitly described the MD methodologies followed. In addition, we have provided the simulation trajectories and starting models as Supplementary Data Files.

In the description of the initial MD refinement experiments we have added, at line 697 “with position restraints on all backbone atoms using a restraint weight of $10 \text{ kcal}\cdot\text{mol}^{-1}\cdot\text{\AA}^{-2}$ and a production run time of 3 ns. Interactions that were observed for at least half of the simulation time were included in the pseudo-atomic model.”

In the subsequent description of the new, longer and explicit, MD simulations we have performed following your suggestions, we have added an additional methods section under the subheading 'Molecular dynamics simulations' at line 721 that reads:

"We originally used implicit solvent MD simulations, of short production run times, to establish interactions that may be occurring at the actomyosin interface. To test if the side chain interactions observed within our primed and postPS actomyosin models at the actin-loop2 and actin-helixW interfaces persist over longer times when subjected to thermal fluctuations, we performed longer MD simulations (> 100 ns; wall clock time of ~5 days), with three independent replicates in explicit solvent conditions for each model. All simulations were run in AMBER16^{54,55} (GPU version PMEMD-CUDA) on all atom systems parameterised with the charmm36m forcefield⁵⁶ and built using CHARMM-GUI⁵⁷. The primed and postPS actomyosin structures were prepared for molecular dynamics simulations using the CHARMM-GUI solution builder⁵⁷⁻⁵⁹. The ADP present in both structures and the Pi present in the primed structure were parameterised from the CHARMM-GUI library with the Pi built as H₂PO₄. All N-termini present in the models were acetylated and C-termini were made neutrally charged. A TIP3P octahedral water box (see Extended Data Fig. 14) was built for each structure using 25mM KCl, 38 mM potassium acetate and 2 mM MgCl₂ (pH 7.0) with a diameter of 186 Å for primed and 172 Å for postPS actomyosin. Each system had a total atom count of 468,219 and 369,572 for primed and postPS respectively. The simulations were then run at a constant temperature of 300 K and constant pressure ensemble (NPT). Simulations were run with position restraints on all backbone atoms using a restraint weight of 5 kcal·mol⁻¹·Å⁻², allowing side chains to move. Each condition was run in triplicate using a different random seed to initiate the starting velocities. Trajectories (see Supplementary Videos 5 & 6 & Supplementary Data Files) were analysed using Chimera and distance plots for atom-atom interaction distances were prepared using GraphPad Prism (Extended Data Fig. 6). The % of time that interacting atoms were within H-bonding distance (3.3Å) was outputted (Extended Data Table 3)."

- 2) Although GBSA simulations are likely to explore more conformational space than normal explicit solvent simulations with similar time spans, due to absence of solvent drag, 3 nanosecond simulations seem to me as too limited for obtaining reliable conclusions. I would prefer to see longer simulations, with independent replicates (3, at least) and, possibly in explicit solvent conditions. The statistics about the interactions (the claimed 50% of these) should be taken by analysing the replicates and not by analysing one sole simulation, which is known to be a significant source of errors and substandard in today's computational capabilities.

We thank the reviewer for these comments. As suggested, we have now performed longer simulations (> 100 ns), with 3 independent replicates, in explicit solvent conditions. We then analysed the % interaction time across all replicates. We find that the interactions we proposed in our models that had not been previously observed or adequately supported by EM density, namely those at the interface between the actin N-terminus and myosin loop2/helixW, were largely maintained during the MD simulations. We are grateful to the referee for suggesting that we perform these experiments as they have strengthened our interpretation of the models.

- 3) One would even improve the analysis further by allowing protein backbone flexibility, in areas where the experimental data does not warrant a proper description of the main fold. But this is debatable.

The experimental data allows accurate fitting of the protein backbone except in the flexible loop regions. These loops have been ill-defined, even in the highest resolution previous structures of the actin-myosin complex, indicating that they are highly disordered. Whilst we agree that it could be an interesting project to simulate the dynamics of loop2 and its potential interactions, such a study is beyond the scope of this work.

- 4) In the answer to reviewer #1, the authors claim that their MD refinement has been used before. While the validity of a method does not rely on its prior use, the authors are not consistent in the defence of their methodologies. They quote the work by Scarff et al. 2020 (the reference quoting the methodology), Bunduc et al. 2021 (where only ISOLDE was used), Dutta et al. 2023 (where, as far I could perceive, MD refinement in the context of cryoEM modelling is used; like in many experimental structural biology methodologies) and Pronker et al. 2023 (where free (with some parts restrained) explicit solvent MD at microsecond scales was employed). Therefore, the arguments towards the use of their methodology are misleading.

We apologise that we were unclear with our response to the reviewer #1 here. Our intention was just to demonstrate to the referee that MD methodologies are being used extensively in the field to refine models from cryoEM data, not to support the use of a specific methodology. We have thus amended this as follows to read at line 132, "To enable detailed interpretation of the cryoEM density maps, we followed current practice²¹⁻²⁴ to create pseudo-atomic models by flexibly fitting available high-resolution structures into the maps, performing homology modelling and refinement and using molecular dynamics (MD) simulation *approaches* to explore side chain interactions at contact interfaces".

In conclusion, I think the work done by the authors has merit overall, but they should repeat the MD "refinement" following what I suggest above, specially the question of independent, longer time-spanning replicates, to infer about the statistical significance of the important interactions described. If possible, explicit solvent simulations should be used.

We thank the referee for acknowledging the merit of our work. As suggested, we have augmented the MD refinement, following this reviewer's suggestions above, performing independent replicates over longer timespans, to determine the statistical significance of the important interactions we describe. Crucially, these extra experiments have added strength to our models, by showing that the interactions we proposed are each largely maintained during the simulations.

Manuscript 2024-01-00287C

Klebl et al. Swinging lever mechanism of myosin directly shown by time-resolved cryoEM

Dear Florian,

We are delighted that you can now offer to publish our manuscript. We can now confirm we have adhered to all the formatting requirements listed in your response letter and specifically respond to each of the numbered/highlighted points below as well as the reviewers' final comments, with our responses in blue.

1: Please organise your article file as follows: Title & front matter, summary, main text, references, methods, data (&code) availability statement, additional references (with continuous numbering), acknowledgements, author contributions, additional information, figure legends, extended data figure legends.

Response: We have now reorganised the manuscript as required.

2: Please submit a revised title within 75 characters (including spaces) that is free of any punctuation marks like colons, exclamation marks, full stops or speech marks.

Response: We have now revised the title to;

Swinging lever mechanism of myosin directly shown by time-resolved cryoEM (73 characters)

3: Please reduce the Abstract to 230 words or less. Currently there are 260 words.

Response: We have now reduced the word count to 230 words and the revised abstract is below.

“Myosins produce force and movement in cells through interactions with F-actin¹. Generation of movement is thought to arise through actin-catalysed conversion of myosin from an ATP-generated primed (pre-powerstroke) state to a post-powerstroke state, exemplified by myosin lever swing^{2,3}. However, the initial, primed actomyosin state has never been observed, and the mechanism by which actin catalyses myosin ATPase activity is unclear. To address these issues, we performed time-resolved cryoEM⁴ of a myosin-5 mutant having slow hydrolysis product release^{5,6}. Primed actomyosin was predominantly captured 10 ms after mixing primed myosin with F-actin, whereas post-powerstroke actomyosin predominated at 120 ms, with no abundant intermediate states detected. For detailed interpretation, cryoEM maps were fitted with pseudo-atomic models. Small but critical changes accompany the primed motor binding to actin through its lower 50 kDa subdomain, with the actin-binding cleft open and phosphate release prohibited. N-terminal actin interactions with myosin promote rotation of the upper 50 kDa subdomain, closing the actin-binding cleft, and

enabling phosphate release. Formation of upper 50 kDa subdomain interactions with actin create the strong-binding interface needed for effective force production. The myosin-5 lever swings through 93°, predominantly along the actin axis, with little twisting. The magnitude of lever swing matches the typical step length of myosin-5 along actin⁷. These time-resolved structures demonstrate the swinging lever mechanism, elucidate structural transitions of the powerstroke, and resolve decades of conjecture on how myosins generate movement.”

4: Please create a separate reference list for the methods with continuous numbering (i.e. do not start at 1 again).

Response: Done as requested.

5: Please remove the main figures from the article file and re-supply them individually in an acceptable format such as EPS, AI, PS, PDF, PPT, PSD or XLS (for graphs) with editable vector files.

Response: All figures have now been uploaded as individual files.

6: Extended data figures should be removed from the Supplementary Information file; please re-supply them individually in EPS, JPEG or TIF format. We only allow up to 10 Extended Data items (tables and figures), currently there are 19. We further suggest that you move ED Figures 1 and 2 to SI as sequence alignments do not display well in this format. Also, please consider combining smaller ED figures (e.g. 5) to reduce the number of items further. The cryo-EM reporting table need to remain in the Extended Data, but we recommend that current ED tables 1-3 and 5 are merged into a single table using our template - see 'DATA DEPOSITION' and 'EXTENDED DATA' below for detailed guidance and links to templates. If the number of ED items is still over 10 after the modifications suggested above, please move items to SI.

Response: We have removed extended data figures from the Supplementary Information file as requested and resupplied them individually. We now have 10 extended data items; three tables and seven figures. We have merged the tables and figures where practical as suggested and any remaining ED figures have been transferred to Supplementary Information.

7: Please ensure that the text size in all figures is at least 5 pt Arial.

Response: We have now checked all figures, and they should confirm to being at least 5 pt Arial.

8: SI videos have been cited in the article file but are currently missing.

Response: All SI videos were originally uploaded to the figshare site. We have now uploaded them as separate files.

9: Please reduce subheadings to 40 characters (with spaces) or less.

Response: As requested, we have reduced all subheadings to 40 characters or less

10: Please provide a supplementary information guide (see 'SUPPLEMENTARY INFORMATION' below).

Response: We have amended the supplementary information as requested and confirm that in addition we have provided a SIGuide.docx

Note the SIGuide contains the Supplementary Video legends. Whilst we have provided legends of maximum length 100 words for each Supplementary Video, in addition, we have provided a longer version for Supplementary Video 4 that we believe is needed to add clarity.

11: There are potential third-party rights issues in some figures (i.e. schematics, illustrations). It is your responsibility to obtain the right to use any items (figures, tables, images, videos or text boxes) that are reproduced (or adapted) from material for which you do not hold copyright and to give proper attribution to the creators of that work. This includes work that has previously been published elsewhere, but also templated from e.g. BioRender. Regardless if third-party material is included, please fill out our and submit it with the revised manuscript. If you generated all illustrations yourself, this can remain empty (except for the author / manuscript number fields).

Response: We have checked the manuscript and are not aware of any figures which have been re-used and therefore subject to third-party rights. We have submitted a blank third-party rights form as requested.

12: Because the manuscript contains Molecular Dynamics simulations, we recommend that a Reliability and reproducibility checklist is included with the paper, this can be a Supplementary Table. Please see here for details:

Response: Due to the nature of the MD simulations we conducted, we do not think that this checklist is necessary in this case. All relevant information is included within the manuscript.

13: In the Editorial Policy Checklist, there is mention of custom computer code. We ask authors to make custom code that is necessary to reproduce the analysis publicly available, please see "DATA AND CODE AVAILABILITY STATEMENTS" below for details. In case no custom code was used, please revise the Editorial Policy Checklist.

Response: The custom computer code related to a small piece of code that operated the time-resolved setup. However, we have now considered the use of this code to the community and as it specifically relates to the machine operation (of which we house the only one), the coding language is not universal and was not used for any data processing or analysis, we have removed this statement. We are happy to share the code with any interested parties but do not feel that publishing the code will help the general community.

14: Editorial Policy Checklist, provision of a 'Inclusion & Ethics' statement was confirmed; however, no such statement was observed in the manuscript. Please include this or provide a revised checklist.

Response: We have provided a revised checklist

15: The data deposited to the EMDB with the dataset identifier EMD-50594 is currently not available. Please ensure its public release.

Response: We have contacted the EMDB and the dataset has now been released.

16: In the 'Data presentation' section of the Editorial Policy Checklist, the field "Individual data points are shown when possible, and always for $n \leq 10$ ", the field corresponding to data distribution formats, and the field asking for clearly defined error bars have been marked as N/A; however, it is observed that these fields are relevant to the study. Please consider revising this.

Response: We have revised accordingly

Comment: TRANSPARENT PEER REVIEW:

Response: 'I wish to participate in transparent peer review'

Comment: ORCID inclusion for all authors

Response: All corresponding authors should now have their ORCID ID linked to their Nature account (Prof White 0000-0003-1066-2051, Dr Scarff 0000-0001-6168-0060, & Prof Muench 0000-0001-6869-4414).

Comment: STATISTICS:

Response: We have checked that all our statistics fit within Nature guidelines.

Comment: There were a series of question within the reporting summary that are detailed below:

For sample size we have now confirmed that the number of experimental repeats is stated in figure legends, extended data fig. 1 and supplementary fig. 3, along with stating that each was a distinct sample. The statistical parameters where relevant are noted.

Bayesian analysis has been carried out as part of the RELION and cryoSPARC packages and we have cited the appropriate papers that explains the choice of priors within the default settings.

For the data collection we have now detailed the version number for EPU (2.12) and cryoSPARC (3.2.0). The following was added on the MD simulations "All simulations were run in AMBER16 (GPU version PMEMD-CUDA) on all atom systems parameterised with the charmm36m forcefield and built using CHARMM-GUI. The primed and postPS actomyosin structures were prepared for molecular dynamics simulations using the CHARMM-GUI solution builder".

We have now included information on sample size, data exclusions, replication and randomization and binding.

Comment: LENGTH:

Response: The manuscript length is within the specified limits.

Referees' comments

Referee #3 (Remarks to the Author):

In their re-revised manuscript, the authors have addressed the remaining major concerns identified in the previous round of review. Notably, they have tempered or removed claims about sidechain positioning and subtle rearrangements, peripheral to the main message of the paper, that were weakly supported by the experimental data. They have also modulated their statement about the lack of intermediate states, which I believe now treats the implications of their classification analysis appropriately. The expanded molecular dynamics are also an excellent addition which strengthens their claims about the key new interactions identified in the primed state, the major take-home of this paper.

I believe this is now suitable for publication in Nature, which is an appropriate venue for the exciting and important structure of the actomyosin primed state.

Referee #4 (Remarks to the Author):

The authors of this manuscript revise it to clarify some points raised in the previous referee report and provide new simulations that enrich the work and solidify the conclusions put forward. I am convinced that the manuscript is now ready for acceptance.

Response: We thank the referees wholeheartedly for their time and contributions. We are pleased that through addressing the comments from the previous rounds of review of the manuscript we have enriched our work and solidified our conclusions. We are delighted that the referees now find our work ready for publication in Nature.